# Mitochondrial Rab GAPs govern autophagosome biogenesis during mitophagy

Koji Yamano[1], Adam I Fogel[1], Chunxin Wang[1], Alexander M van der Bliek[2], Richard J Youle[1]*

[1]Biochemistry Section, Surgical Neurology Branch, National Institute of Neurological Disorders and Stroke, National Institutes of Health, Bethesda, United States; [2]Department of Biological Chemistry, David Geffen School of Medicine at University of California, Los Angeles, Los Angeles, United States

**Abstract** Damaged mitochondria can be selectively eliminated by mitophagy. Although two gene products mutated in Parkinson's disease, PINK1, and Parkin have been found to play a central role in triggering mitophagy in mammals, how the pre-autophagosomal isolation membrane selectively and accurately engulfs damaged mitochondria remains unclear. In this study, we demonstrate that TBC1D15, a mitochondrial Rab GTPase-activating protein (Rab-GAP), governs autophagosome biogenesis and morphology downstream of Parkin activation. To constrain autophagosome morphogenesis to that of the cargo, TBC1D15 inhibits Rab7 activity and associates with both the mitochondria through binding Fis1 and the isolation membrane through the interactions with LC3/GABARAP family members. Another TBC family member TBC1D17, also participates in mitophagy and forms homodimers and heterodimers with TBC1D15. These results demonstrate that TBC1D15 and TBC1D17 mediate proper autophagic encapsulation of mitochondria by regulating Rab7 activity at the interface between mitochondria and isolation membranes.

*For correspondence: youle@helix.nih.gov

## Introduction

Autophagosomes enclose seemingly random portions of the cytoplasm to supply nutrients during starvation or they can specifically engulf cellular debris to maintain quality control (*Mizushima et al., 2011*). Although these two pathways share a number of core biochemical steps, the morphology of the isolation membranes can differ greatly. Cup-shaped isolation membranes surround cytosol and organelles upon starvation (*Baba et al., 1994*). However, when large aggregates of debris accumulate or during xenophagy (*Kageyama et al., 2011*), autophagosomes form locally on the surface of the substrate and grow in contour with the often irregularly shaped structures that can be much larger than the membrane cups formed during starvation. Furthermore, for large aggregates of mitochondria, several autophagosomes may form simultaneously on different sides of the cargo and fuse together to engulf them (*Yoshii et al., 2011*). Ubiquitinated protein aggregates may be tagged by LIR (LC3-interacting region) domain-containing proteins that bind LC3 family proteins on isolation membranes to help recruit autophagosomes to the debris (*Birgisdottir et al., 2013*). However, what links isolation membrane expansion to cargo size is unknown.

We examined autophagosome morphogenesis during Parkin-mediated mitophagy (*Youle and Narendra, 2011*) where multiple mitochondria can be engulfed in large aggregates (*Narendra et al., 2010a*; *Yoshii et al., 2011*). A broad range of mitochondrial outer membrane proteins become ubiquitinated by Parkin (*Chan et al., 2011*; *Sarraf et al., 2013*), which appears to trigger the recruitment of autophagy-related (Atg) proteins to mitochondria and activate autophagosome assembly

**eLife digest** Parkinson disease is a common degenerative brain disorder that causes tremors, muscle stiffening, and slowing down of movement. Scientists believe that these symptoms are caused by a progressive loss of brain cells called dopaminergic neurons, which help regulate movement. Most cases have no obvious genetic cause, but around 15% of people with the disease have a close relative who also has the disease, and mutations in the genes encoding two proteins—PINK1 and Parkin—have been identified as prime suspects in familial Parkinson disease.

These proteins help to eliminate damaged mitochondria from cells. In addition to producing the energy that cells need to function, mitochondria also help to trigger cell death. Pesticides and other chemicals linked to non-familial cases of Parkinson disease also damage mitochondria. Taken together, this evidence suggests that the accumulation of damaged mitochondria may contribute to the excessive loss of dopaminergic neurons that is seen in both forms of the disease.

Yamano et al. provide new details on the ways that autophagosomes—structures that help cells to recycle nutrients and remove debris—destroy mitochondria. Previous studies have shown that when a mitochondrion is damaged, PINK1 sends a signal to Parkin, which then helps to recruit the proteins that are needed to form an autophagosome around the damaged mitochondrion. However, the identity of the proteins that guide the formation of the autophagosome remained a mystery.

Yamano et al. have now identified two of these proteins and helped to explain their specific roles in the assembly of autophagosomes. The two proteins, which are called TBC1D15 and TBC1D17, are both GAP proteins, which are well known for their role in deactivating enzymes called RAB GTPases. Yamano et al. show that TBC1D15 binds to the damaged mitochondrion and also to the autophagosome as it grows around the mitochondrion. TBC1D15 also inhibits the action of an enzyme called Rab7 to prevent excessive growth of the autophagosome. TBC1D17 has a similar role.

The work of Yamano et al. indicates that Parkin activates Rab7, perhaps by placing chains of a protein called ubiquitin on mitochondria, which would mean that an unexpected new step in this pathway remains to be discovered. Understanding how Parkin activates Rab7 could help identify new targets for drugs that might treat Parkinson disease.

(*Itakura et al., 2012*). Although mitophagy is accomplished in part by canonical autophagy gene products, how autophagosomes selectively recognize and assemble around damaged mitochondria remains unclear. So far, no mitochondrial proteins have been identified to mediate mitophagy downstream of Parkin.

TBC1D15 has been shown to bind mitochondria through Fis1 (*Onoue et al., 2013*), to bind the Atg8 family member GABARAP (*Behrends et al., 2010*), and has been found in a screen of Parkin substrates (*Sarraf et al., 2013*). TBC1D15 is one of the TBC (Tre-2/Bub2/Cdc16) family proteins that function as Rab GTPase-activating proteins (Rab-GAPs) (*Fukuda, 2011*; *Frasa et al., 2012*). The membrane trafficking activity of Rab proteins is increased by guanine nucleotide exchange factors (GEFs), which accelerate the exchange of GDP for GTP and is decreased by GAPs, which facilitate the Rab GTPase activity.

Fis1, conserved from yeast to mammals, is C-terminally anchored in the mitochondrial outer membrane (*Mozdy et al., 2000*) with an N-terminal tetratricopeptide-repeat (TPR) domain exposed to the cytosol (*Suzuki et al., 2003*). Although mammalian Fis1 had been thought to be a mitochondrial fission factor in the same way as yeast Fis1, Fis1 knock out mammalian cells do not display a defect in mitochondrial fission (*Otera et al., 2010*). Also in contrast to yeast, mammalian Drp1 recruitment to mitochondria requires Mff, MIEF1/MiD51, and/or MiD49 (*Otera et al., 2010*; *Palmer et al., 2011*; *Zhao et al., 2011*). Interestingly, the Fis1 null worm and Fis1 knock out mammalian cells have excessive LC3 accumulation following stress induced by mitochondrial inhibitors such as antimycin A in a PINK1-dependent manner (*Shen et al., 2014*) suggesting Fis1 may have a role in mitophagy.

In this study, we show that TBC1D15 and Fis1 act in concert to control autophagosome morphology during Parkin-mediated mitophagy but not during starvation-induced autophagy. This step is dependent on the small GTPase Rab7. We also demonstrate that TBC1D15 must bind LC3 homologue proteins as well as Fis1 to coordinate Rab7 activity to shape the nascent autophagosome isolation membrane providing mechanistic insight into autophagosome morphogenesis during mitophagy.

## Results

### Characterization of *TBC1D15−/−* cells

We recently found that Fis1 null *C.elegans* and mammalian cells display aberrant LC3 accumulation (*Shen et al., 2014*). However, the molecular mechanism underlying the association between LC3 accumulation and the loss of Fis1 remains unclear. As TBC1D15 binds to Fis1 (*Onoue et al., 2013*) and is a Rab-GAP potentially involved in autophagy (*Behrends et al., 2010*), we made a *TBC1D15* gene knock out (KO) cell line using TALENs (Transcription activator-like effector nucleases [*Gaj et al., 2013*]) and compared the LC3 accumulation phenotype with that of *FIS1−/−* cells. We designed TALEN binding pairs that target exon 9 of the *TBC1D15* gene because it is shared by all conceivable TBC1D15 splicing isoforms. One *TBC1D15−/−* clone harbors a 14-bp deletion in one allele and a large deletion in the other allele, both of which are frame-shifting and would cause mRNA decay (*Figure 1—figure supplement 1*). We confirmed the knock out of TBC1D15 expression by immunoblotting (*Figure 1A*). As TBC1D15 was previously reported to mediate mitochondrial fission and bind to Fis1 (*Onoue et al., 2013*), we checked the expression level of several proteins that have been previously linked to mitochondrial fission pathways in the *TBC1D15−/−* cells as well as in *FIS1−/−*, *MFF−/−*, *DRP1−/−* and the corresponding WT HCT116 cells (*Figure 1A*). Although each KO cell line has complete deletion of the target protein, expression and/or stability of the other fission-related proteins was not affected; *FIS1−/−* and *TBC1D15−/−* cells possess normal levels of TBC1D15 and Fis1, respectively.

It has been reported that several mitochondrial fission components such as Mff, Drp1, and Fis1 are localized not only on mitochondrial membranes but also on peroxisomal membranes (*Li and Gould, 2003*; *Kobayashi et al., 2007*; *Koch and Brocard, 2012*). We compared the mitochondrial and peroxisomal morphology of *TBC1D15−/−* cells with that of *FIS1−/−*, *MFF−/−*, and *DRP−/−* cells. In agreement with previous work (*Otera et al., 2010*), mitochondrial morphology, as well as peroxisomal morphology in *FIS1−/−* cells, was similar to that of WT cells (*Figure 1B*). In contrast to moderately elongated mitochondria in *TBC1D15* siRNA-treated cells reported previously (*Onoue et al., 2013*), complete depletion of TBC1D15 by knock out resulted in no obvious mitochondrial morphology changes (*Figure 1B,C*). Furthermore, peroxisome shape in *TBC1D15−/−* cells was also indistinguishable from that of WT cells (*Figure 1B*). In sharp contrast, *MFF−/−* and *DRP−/−* cells display elongated mitochondria and peroxisomes (*Figure 1B*), indicating that Mff and Drp1 play important roles in regulating both mitochondria and peroxisome morphology, consistent with the previous findings (*Smirnova et al., 1998*; *Koch et al., 2003*; *Gandre-Babbe and van der Bliek, 2008*). Quantification of mitochondrial morphology demonstrated that the fission defect in *MFF−/−* cells is similar to, but not as strong as, that caused by *DRP1* deletion (*Figure 1C*). Therefore, it appears that both TBC1D15 and Fis1 are dispensable for mitochondrial and peroxisomal fission in human cells.

### LC3 accumulation is induced excessively in *TBC1D15−/−* cells during Parkin-mediated mitophagy

*FIS1−/−* cells accumulate LC3 during Parkin-mediated mitophagy (*Shen et al., 2014*). To assess autophagosomes in the absence of TBC1D15, we made stable cell lines expressing YFP-LC3 and mCherry-Parkin and treated the cells with valinomycin, a potassium ionophore that dissipates mitochondrial inner membrane potential, to trigger Parkin-mediated mitophagy. 3 hr of valinomycin treatment to depolarize mitochondria-stimulated Parkin translocation onto mitochondria with similar efficiencies in all cell lines tested (*Figure 2A,B*). WT cells have several small dots, crescent-shaped, or spherical YFP-LC3 puncta representing isolation membranes and autophagosomes, respectively, that were observed on or near fragmented mitochondria, consistent with previous observations (*Figure 2A* and *Narendra et al., 2008*). However, upon mitophagy induction in *FIS1−/−* cells, YFP-LC3 accumulates excessively in foci, which often appears interconnected with one another, localized on or near mitochondria (*Figure 2A,C* and *Figure 2—figure supplement 2*; *Shen et al., 2014*). Interestingly, *TBC1D15−/−* cells display expanded LC3-labeled structures very similar to those in *FIS1−/−* cells suggesting that Fis1 and TBC1D15 are involved at the same step in autophagy (*Figure 2A,C* and *Figure 2—figure supplement 2*). This phenotype requires both Parkin (*Figure 2—figure supplement 3A*) and PINK1 (*Figure 2—figure supplement 3B*) expression and the loss of the inner mitochondrial membrane potential (*Figure 2—figure supplement 3C*), indicating that it occurs upon mitophagy induction downstream of PINK1-induced Parkin translocation to mitochondria. Expression of N-terminally 3×FLAG-tagged Fis1 rescued the LC3 accumulation in *FIS1−/−* cells (*Figure 2—figure supplement 4*),

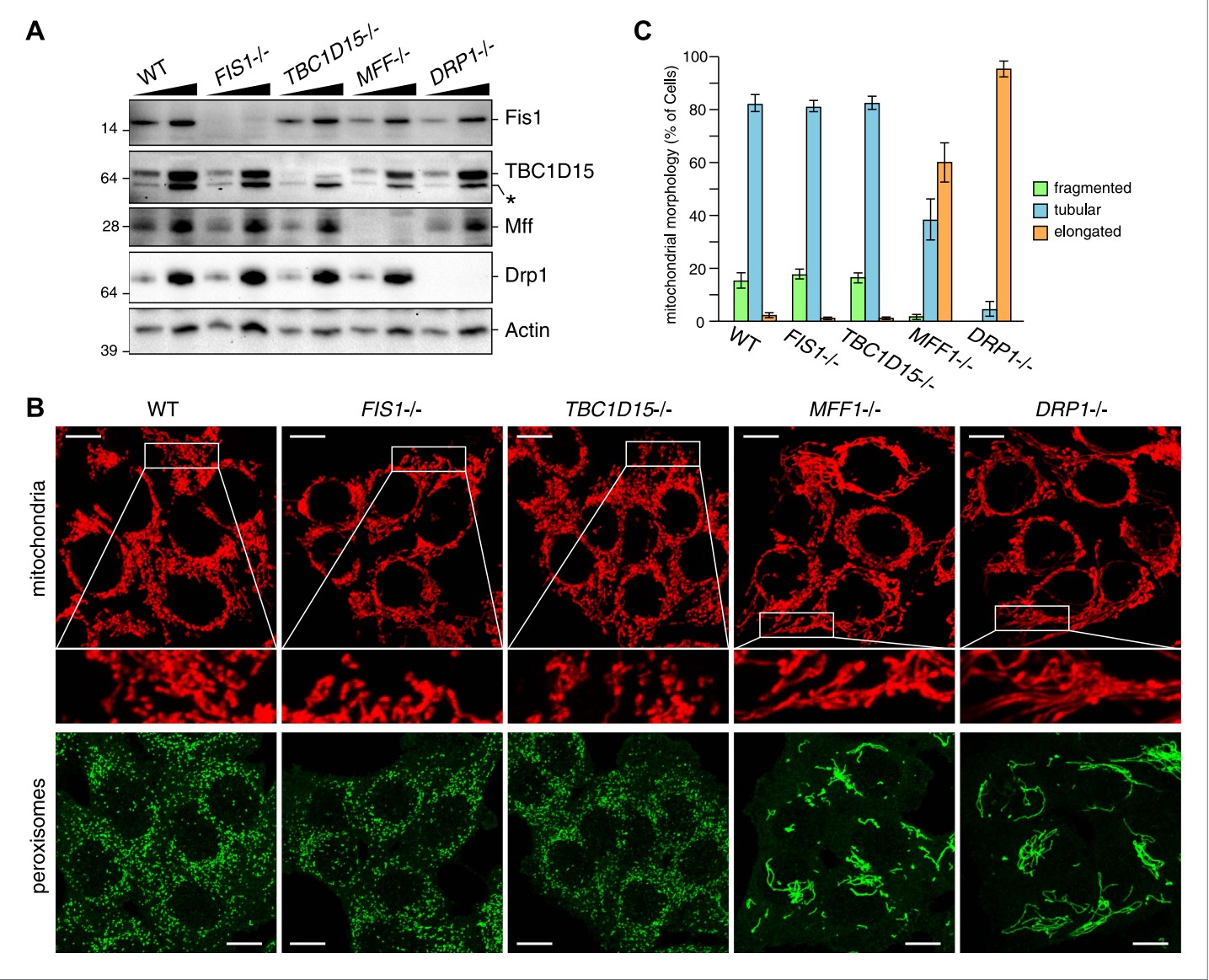

**Figure 1**. TBC1D15 is dispensable for mitochondria and peroxisome morphologies. (**A**) Total cell lysates prepared from the indicated HCT116 cell lines were analyzed by immunoblotting. For comparison, different amounts of proteins (1:3 ratio) were applied. An asterisk indicated non-specific crossreactive bands. (**B**) The indicated cell lines were analyzed by immunofluorescence microscopy using anti-Cytochrome c antibody for mitochondria and anti-PMP70 antibody for peroxisome staining. Images are displayed as z-stacks of 6 confocal slices. Magnified images are also shown for mitochondrial morphologies. Scale bars, 10 µm. (**C**) Quantification of mitochondrial morphologies in (**B**). Percentages of cells harboring fragmented, tubular, or elongated mitochondria are shown. Tubular and elongated denote normal tubular mitochondria seen in WT cells and highly connected mitochondrial network, respectively. The error bars represent ±SD from three independent replicates. Over 50 cells were counted in each of three replicate wells.

The following figure supplements are available for figure 1:

**Figure supplement 1**. TBC1D15 gene knock out by TALENs.

confirming the effect was caused by the loss of Fis1. Similarly, N-terminally HA-tagged WT TBC1D15 expressed in *TBC1D15−/−* cells reverted the excessive LC3 accumulation to that of WT after mitophagy induction (*Figure 2D,E*).

To clarify the nature of LC3 accumulation in *TBC1D15−/−* cells in more detail, we conducted immunoelectron microscopy. Valinomycin treatment for 3 hr caused the formation of preautophagomes with closely apposed membranes that were labeled with gold particles attached to YFP-LC3 near mitochondria in WT cells (*Figure 2F*). Although *TBC1D15−/−* cells also have similar cup-shaped membrane

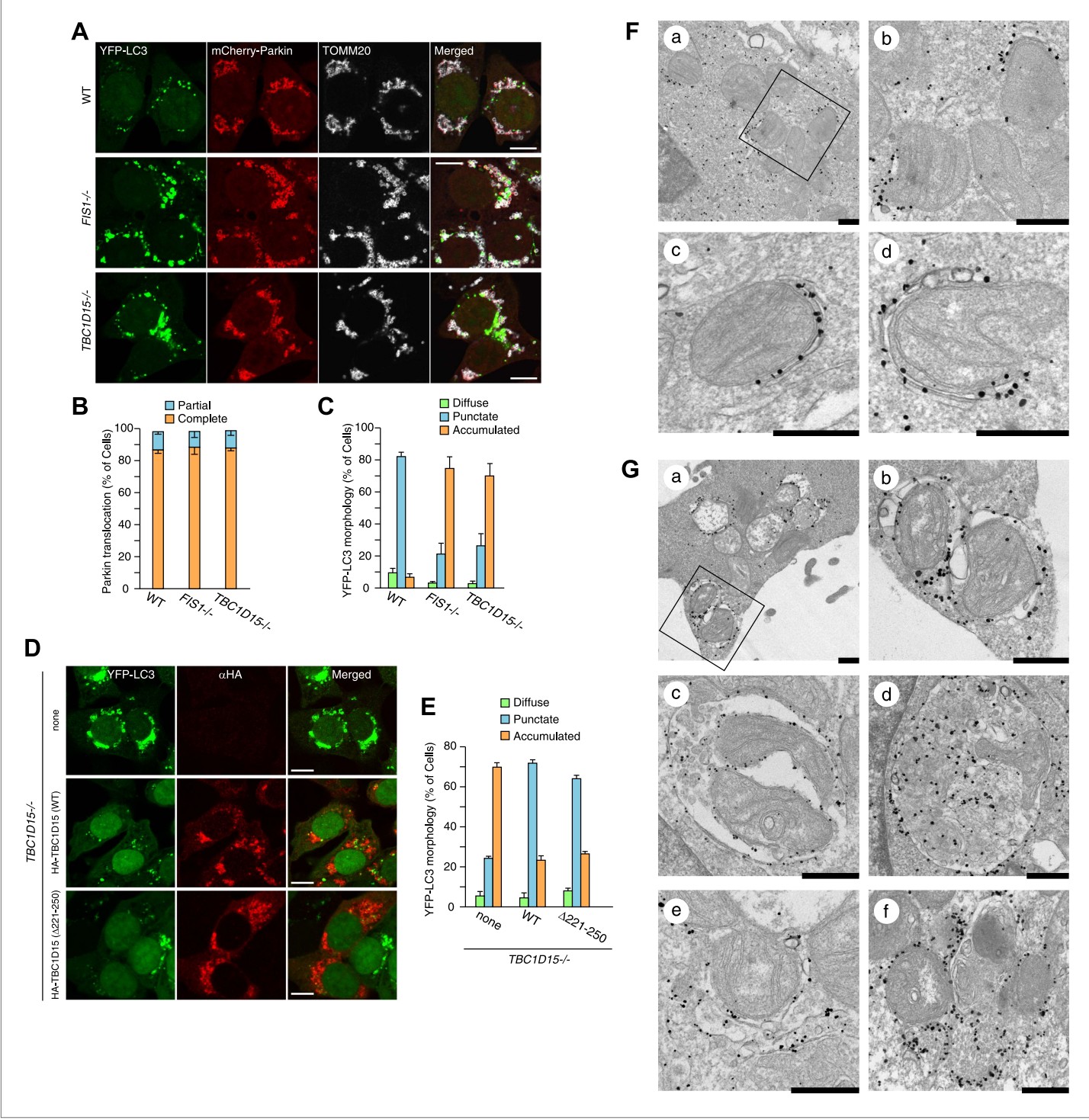

**Figure 2**. Loss of Fis1 or TBC1D15 causes LC3 accumulation during mitophagy. (**A**) The indicated cell lines stably expressing YFP-LC3 and mCherry-Parkin were treated with valinomycin for 3 hr and subjected to confocal immunofluorescence microscopy with anti-TOMM20 antibody. Scale bars, 10 μm. (**B**) Quantification of mCherry-Parkin translocation to mitochondria after 3 hr of valinomycin treatment. Partial or complete translocation to mitochondria in each cell was scored as separate phenotypes. Partial and complete denote that Parkin translocates to some of or all mitochondria, respectively. The error bars represent ±SD from three independent experiments. Over 100 cells were counted in each of three separate wells. (**C**) YFP-LC3 morphologies in (**A**) were quantified. Percentages of cells harboring diffuse, punctate or accumulated YFP-LC3 are shown. The error bars represent ±SD from three independent replicates. Over 100 cells were counted in each replicate. For the criteria of LC3 morphology, see *Figure 2—figure supplement 1*. (**D**) YFP-LC3 and mCherry-Parkin stably expressing *TBC1D15−/−* cells in the absence or presence of HA-tagged TBC1D15 WT or Δ221-250 mutant were
*Figure 2. Continued on next page*

*Figure 2. Continued*

treated with valinomycin for 3 hr. Cells were subjected to immunofluorescence microscopy with anti-HA antibody. Scale bars, 10 μm. (**E**) The YFP-LC3 morphology of cells in (**D**) was quantified. The error bars represent ±SD from three independent replicates. Over 50 cells were counted in each well. (**F** and **G**) YFP-LC3 and mCherry-Parkin stably expressing WT (**F**) and TBC1D15−/− (**G**) cells were treated with valinomycin for 3 hr and then subjected to immunoelectron microscopy with anti-GFP antibody. The square in panel a shows enlarged areas in panel b Scale bars, 500 nm.

The following figure supplements are available for figure 2:

**Figure supplement 1**. Image examples of different LC3 morphologies.

**Figure supplement 2**. YFP-LC3 accumulation in *FIS1*−/− and *TBC1D15*−/− cells during mitophagy.

**Figure supplement 3**. Excessive LC3 accumulation depends on PINK1/Parkin-mediated mitophagy.

**Figure supplement 4**. LC3 accumulation in *FIS1*−/− cells was rescued by Fis1 re-expression.

**Figure supplement 5**. ULK1 and Atg14 recruitment in WT, *FIS1*−/− and *TBC1D15*−/− cells during mitophagy.

**Figure supplement 6**. Atg16L1 and DFCP1 recruitment in WT, *FIS1*−/− and *TBC1D15*−/− cells during mitophagy.

structures, many of them displayed much thicker lumens than those in WT cells (*Figure 2G*, panel a and b). In addition, larger membrane capsules that contain mitochondria and possibly other membrane organelles were observed only in *TBC1D15* −/− cells (*Figure 2G*, panel c and d). Furthermore, some LC3 was associated with small diameter membrane structures that localize very close to mitochondria (*Figure 2G*, panel e) and may have out of plane contiguity to preautophagosomes (*Figure 2G*, panel e and f). These results strongly suggest that the LC3-labeled structures seen in *TBC1D15*−/− cells are not simply LC3 protein aggregates, but express LC3 associated along membrane bilayers.

We compared the localization of other preautophagosomal marker proteins and proteins involved in autophagy induction among WT, *FIS1*−/−, and *TBC1D15*−/− cells during mitophagy. GFP-ULK1 that functions in the most upstream ATG1/ULK kinase complex and GFP-Atg14 that is an autophagy specific subunit of the class III PI3K complex formed dots and/or cup-shaped structures around mitochondria following valinomycin treatment, although the number of dots did not further increase upon the loss of Fis1 or TBC1D15 (*Figure 2—figure supplement 5*). Moreover, we analyzed the recruitment of endogenous Atg16L1 that is only present on the preautophagosome membrane and GFP-DFCP1 that can serve as an omegasome marker. Although valinomycin induced the formation of both Atg16L1 and DFCP1 dots on mitochondria, neither of them increased further in *FIS1*−/− or *TBC1D15*−/− cells (*Figure 2—figure supplement 6*). Thus, Fis1 and TBC1D15 seem to regulate LC3 accumulation downstream of autophagy initiation.

## LC3-labeled tubule expansion in *TBC1D15*−/− cells occurs along microtubules

To observe autophagosomal structures in thicker optical slices, we reconstructed z-stacked confocal microscopic images comprising six sequential optical x-y sections taken at 0.8 μm z-intervals of cells stably expressing YFP-LC3 and mCherry-Parkin after 3 hr treatment with valinomycin. WT cells treated with valinomycin display perinuclear-clustered fragmented mitochondria and many YFP-LC3 puncta associated with them (*Figure 3A*). In sharp contrast, upon Parkin-induced mitophagy, *TBC1D15*−/− cells display more expanded LC3 accumulation than WT cells and also generate thin LC3-labeled tubular structures interconnected to one another, which are very similar to that seen in *FIS1* −/− cells (*Figure 3A*). These tubular structures were frequently observed near the cell cortex. Live cell imaging showed that the LC3-containing tubules emanating from the LC3 foci are highly mobile (*Video 1*). Immunoelectron microscopy confirmed that long YFP-LC3-labeled tubules observed in *TBC1D15*−/− cells are membrane associated (*Figure 3B*)

Although these LC3-labeled tubules were not associated with ER or actin filaments (*Figure 3—figure supplement 1A,B*), a portion of them tightly colocalized with microtubules stained for Tubulin (*Figure 3C*). To examine whether microtubules are important for generating LC3 tubular structures in *TBC1D15*−/− cells, we added nocodazole, which interferes with the polymerization of microtubules,

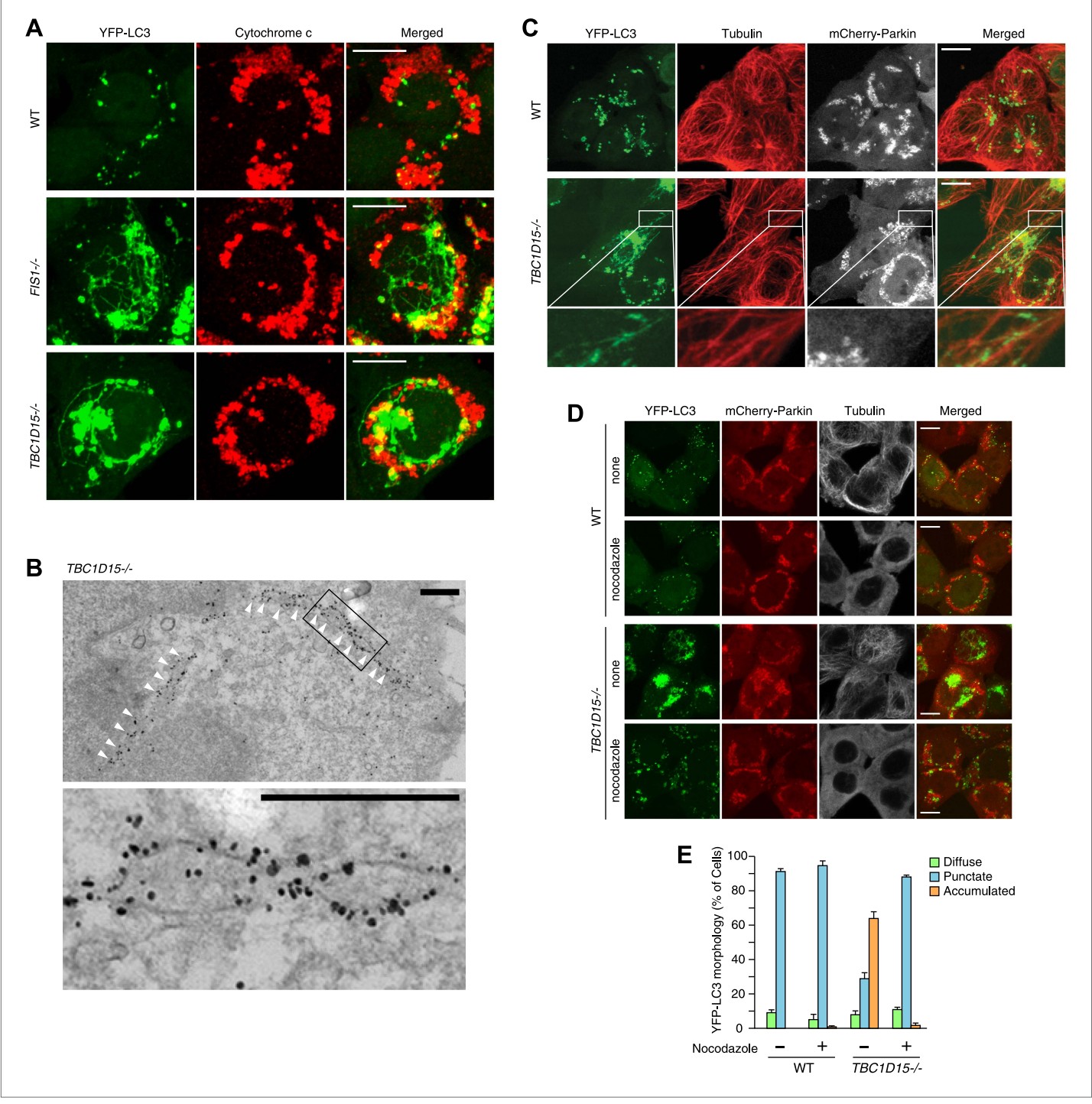

**Figure 3**. Tubular LC3 expands along microtubules. (**A**) The indicated cells stably expressing YFP-LC3 and mCherry-Parkin were treated with valinomycin for 3 hr followed by immunofluorescence microscopy with anti-Cytochrome c antibody. Confocal images were acquired as z-stacks comprising 6 sequential sections with 0.8 μm z-intervals. Scale bars, 10 μm. (**B**) *TBC1D15−/−* cells prepared as in (**A**) were subjected to immunoelectron microscopy with anti-GFP antibody. White arrowheads indicate YFP-LC3-labeled tubules. High magnification image is shown in the lower panel. Scale bars, 500 nm. (**C**) The indicated cells prepared as in (**A**) were subjected to immunostaining with anti-Tubulin antibody. YFP-LC3 and Tubulin staining are merged in the right panels. Magnified images are shown for *TBC1D15−/−* cells. Scale bars, 10 μm. (**D**) The indicated cells stably expressing YFP-LC3 and mCherry-Parkin were treated with valinomycin in the presence or absence of nocodazole for 3 hr. YFP-LC3 and mCherry-Parkin images are merged in the right panels. Scale bars, 10 μm. (**E**) YFP-LC3 morphologies of cells in (**D**) were quantified. The error bars represent ±SD from three independent replicates. Over 50 cells were counted in each well.

*Figure 3. Continued on next page*

*Figure 3. Continued*

The following figure supplements are available for figure 3:

**Figure supplement 1**. Tubular YFP-LC3 morphologies in *TBC1D15−/−* cells.

together with valinomycin to cells for 3 hr. Although WT cells have similar LC3 puncta with or without nocodazole treatment, LC3-labeled tubular structures and LC3 accumulation in *TBC1D15−/−* cells completely disappeared upon nocodazole treatment (*Figure 3D,E*). Thus, intact microtubules are essential for LC3-labeled membrane expansion in *TBC1D15−/−* cells, a step upstream of autophagosome fusion with lysosomes where previous work indicated dynein-facilitated autophagy (*Kimura et al., 2008*). Although some late endosomes/lysosomes colocalize with LC3 (*Figure 3—figure supplement 1C*) consistent with the results of immunoelectron microscopy (*Figure 2G*, panel f), the morphologies of lysosomes and the Golgi apparatus were similar between WT and *TBC1D15−/−* cells (*Figure 3—figure supplement 1C,D*).

As the excessive LC3 accumulation in the absence of TBC1D15 indicates either an augmentation in autophagy induction or a defect in autophagosomal flux, we asked whether the loss of TBC1D15 alters the rate of Parkin-mediated mitophagy. For this purpose, YFP-Parkin stably expressing cells were treated with valinomycin for various times and total cell lysates were analyzed by immunoblotting. More endogenous lipidated LC3 (LC3-II) accumulated in *TBC1D15−/−* cells as compared to that of WT cells as also seen in *FIS1−/−* cells (*Figure 4A*). This is consistent with the greater appearance of YFP-LC3 accumulation by microscopic observation (*Figure 2A*). Although Mfn1, a mitochondrial outer membrane protein, was completely degraded within 2 hr in WT, *FIS1−/−* and *TBC1D15−/−* cells, ubiquitination of TOMM20, another outer membrane protein was retarded in *FIS1−/−* and *TBC1D15−/−* cells (*Figure 4A*; TOMM20 asterisk). In WT cells, PINK1 is barely detectable under normal growing conditions because of the degradation through N-end rule pathway (*Yamano and Youle, 2013*), whereas it accumulates on the outer membrane upon addition of valinomycin (*Figure 4A*) as seen also with CCCP (*Matsuda et al., 2010*; *Vives-Bauza et al., 2010*; *Narendra et al., 2010b*). The accumulated PINK1 starts to decrease after 8 hr of valinomycin likely as mitochondria are eliminated by Parkin-mediated mitophagy. On the other hand, accumulated PINK1 in *FIS1−/−* or *TBC1D15−/−* was retained longer than in WT cells (*Figure 4A*), suggesting, in conjunction with the delayed TOMM20 degradation, that autophagic clearance and/or proteasomal degradation is delayed in *FIS1−/−* or *TBC1D15−/−* cells. The rate of degradation of TOMM20, TIMM23, and mitochondrial HSP60 in *FIS1−/−* or *TBC1D15−/−* cells, however, was not different as compared to that of WT cells at the 8-hr time points (*Figure 4A*). On the other hand, when cells were treated with valinomycin for longer times up to 40 hr, most of the TOMM20 was degraded and protein levels of the matrix HSP60 and the mitochondrial DNA-encoded COXII (MT-COXII) decreased in WT cells, whereas they were retained in *FIS1−/−* and *TBC1D15−/−* cells (*Figure 4B,C*). These results indicate that the Fis1-TBC1D15 pair is not essential for mitophagy but the deletion of either impedes clearance of damaged mitochondria, despite the greater magnitude of LC3 accumulation following Parkin translocation. We also found that, although Fis1 is degraded during mitophagy, TBC1D15 escapes the degradation (*Figure 4D*).

## Role of mitochondrial fission and starvation on autophagosome formation induced by the absence of Fis1 or TBC1D15

Mitochondrial fission has been reported to be involved in mitophagy (*Twig et al., 2008*;

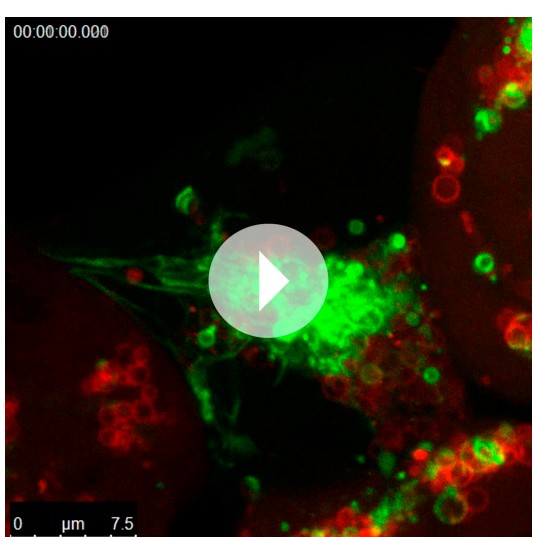

**Video 1**. Tubular LC3 movement in *TBC1D15−/−* cells. *TBC1D15−/−* cells stably expressing YFP-LC3 (green) and mCherry-Parkin (red) were treated with valinomycin for 3 hr and then took z-stack pictures.

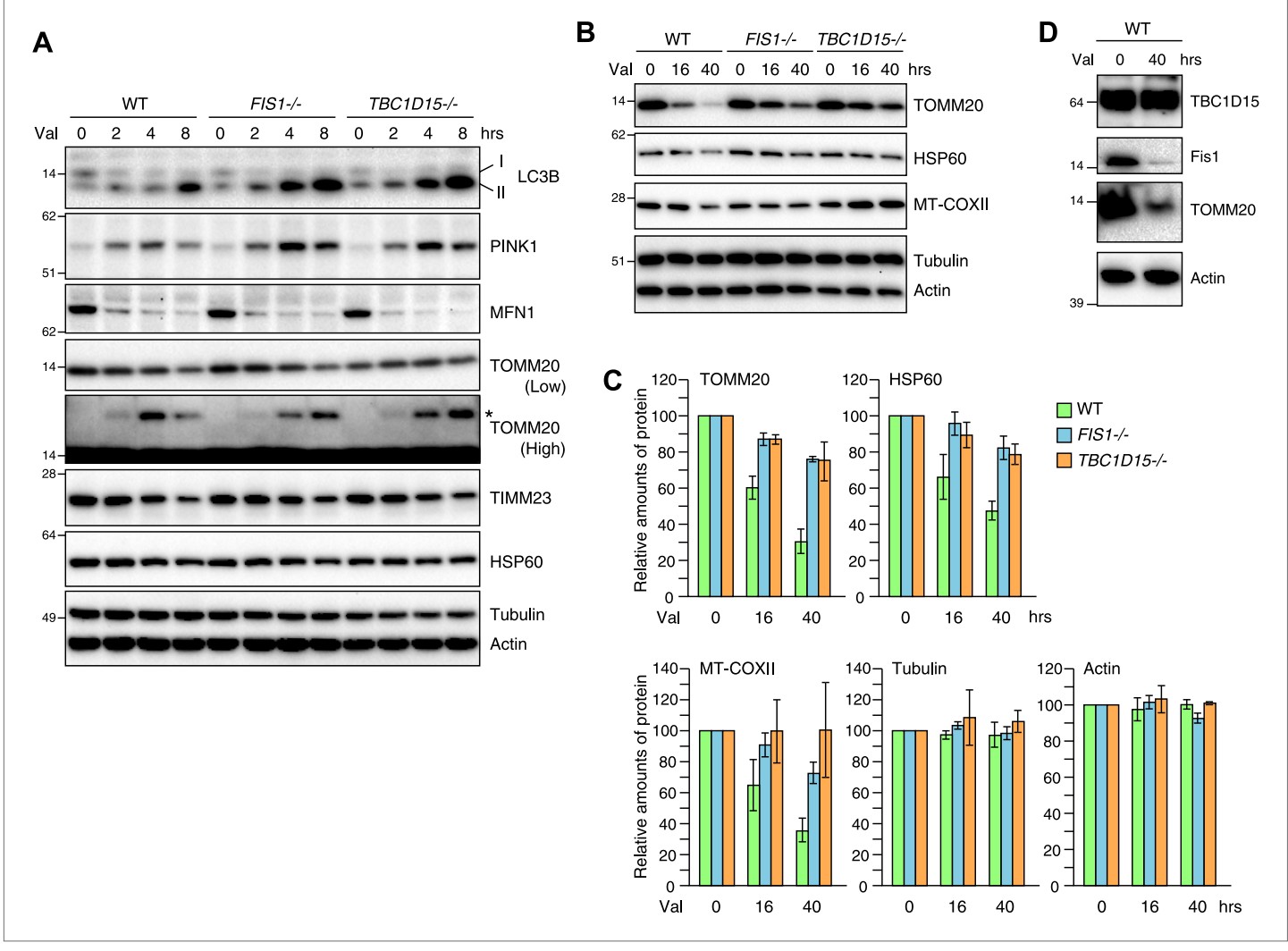

**Figure 4**. Loss of Fis1 or TBC1D15 impedes clearance of damaged mitochondria. (**A** and **B**) YFP-Parkin stably expressing cells were treated with valinomycin for indicated times. Total cell lysates were subjected to immunoblotting. I and II denote cytosolic and lipidated LC3B, respectively. An asterisk indicates ubiquitinated TOMM20. (**C**) Indicated protein amounts as in (**B**) were quantified. The amount of protein without valinomycin treatment was set to 100%. The error bars represent ±SD from three independent experiments. (**D**) YFP-Parkin stably expressing WT HCT116 cells were treated with or without valinomycin for 40 hr. Total cell lysates were analyzed by immunoblotting.

*Tanaka et al., 2010*; *Mao et al., 2013*). Therefore, we assessed if fission defects generally cause LC3 accumulation. The excessively long mitochondria found in *MFF−/−* or *DRP−/−* cells are converted from an elongated tubular network to larger, more rounded, or bubble-like structures relative to WT cells upon 3 hr valinomycin treatment (*Figure 5A*). However, Parkin translocation and mitochondrial colocalization with YFP-LC3 in *MFF−/−* and *DRP1−/−* cells were similar to those in WT cells (*Figure 5A–C*) without the LC3 accumulation and/or tubulation seen in *FIS1−/−* and *TBC1D15−/−* cells (*Figure 2A* and *Figure 3A*). These results suggest that LC3 accumulation is a unique feature following the loss of Fis1 or TBC1D15 and is not caused by defects in mitochondrial fission, consistent with the finding that Fis1 and TBC1D15 do not appear to regulate mitochondrial morphology (*Figure 1B*).

We also investigated the effect of loss of Fis1 or TBC1D15 on starvation-induced autophagy by incubating cells in media lacking amino acids. Upon starvation, the number of YFP-LC3 puncta greatly increased, but indistinguishably between WT, *FIS1−/−*, and *TBC1D15−/−* cells (*Figure 5D,E*). Additionally, immunoblotting of endogenous LC3 demonstrated that although more of the lipidated LC3 was generated upon starvation, the amount was not further increased by the depletion of Fis1

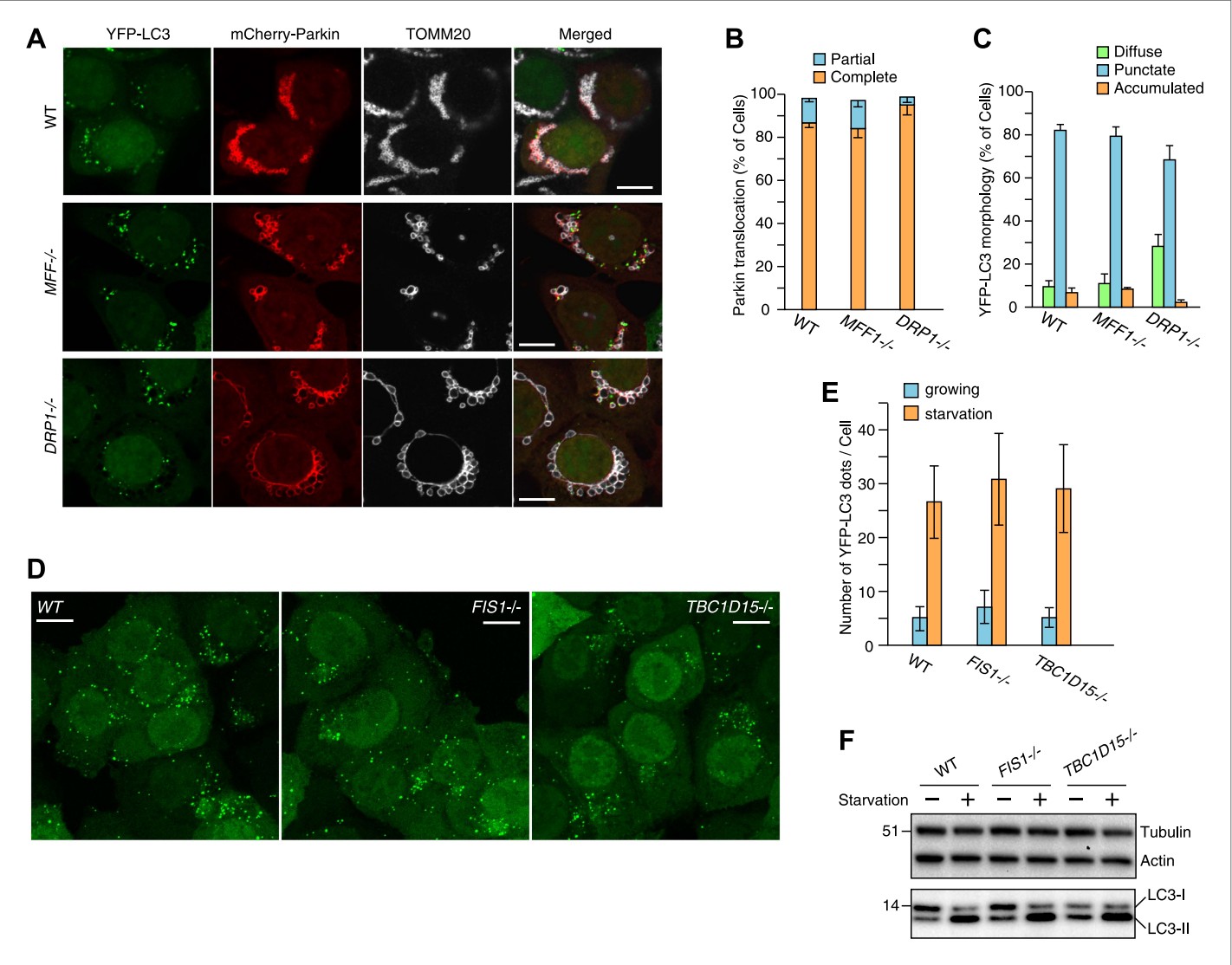

**Figure 5**. Effect of mitochondrial fission and starvation induction on LC3 accumulation in WT, *FIS1−/−* and *TBC1D15−/−* cells. (**A**) WT, *MFF−/−* or *DRP1−/−* cells stably expressing YFP-LC3 and mCherry-Parkin were treated with valinomycin for 3 hr and subjected to immunofluorescence microscopy with anti-TOMM20 antibody. Scale bars, 10 μm. (**B**) Quantification of mCherry-Parkin translocation to mitochondria after 3 hr of valinomycin treatment. Partial and complete denote that Parkin translocates to some of and all mitochondria, respectively. The error bars represent ±SD from three independent replicates. Over 50 cells were counted in each replicate. (**C**) YFP-LC3 morphologies of cells in (**A**) were quantified. Percentages of cells harboring diffuse, punctate or accumulated YFP-LC3 are shown. The error bars represent ±SD from three independent replicates. Over 100 cells were counted in each replicate. (**D**) WT, *FIS1−/−*, and *TBC1D15−/−* cells stably expressing YFP-LC3 were grown in starvation media. Z-stacks of confocal images are shown. Scale bars, 20 μm. (**E**) The number of YFP-LC3 dots in cells under growth or starvation conditions was quantified. The error bars represent ±SD from three independent replicates. Over 50 cells were counted in each well. (**F**) Total cell lysates from cells grown in normal or starvation media for 6 hr were subjected to immunoblotting. LC3-I and LC3-II denote cytosolic and lipidated LC3B, respectively.

or TBC1D15 (*Figure 5F*). Therefore, TBC1D15 and Fis1, which form a complex on mitochondria, seem to govern autophagosome biogenesis specifically during mitophagy.

## Identification of TBC1D17 as a Fis1 and TBC1D15 interacting protein

To investigate whether the Fis1 binding of TBC1D15 is important for LC3 accumulation, we introduced into *TBC1D15−/−* cells HA-tagged TBC1D15 lacking amino acids 221-250 (HA-TBC1D15 Δ221-250), which was reported to be deficient in Fis1 binding (*Onoue et al., 2013*). Unexpectedly, mitochondrial localization of TBC1D15 Δ221-250 was observed under normal growing conditions (*Figure 6A*) and

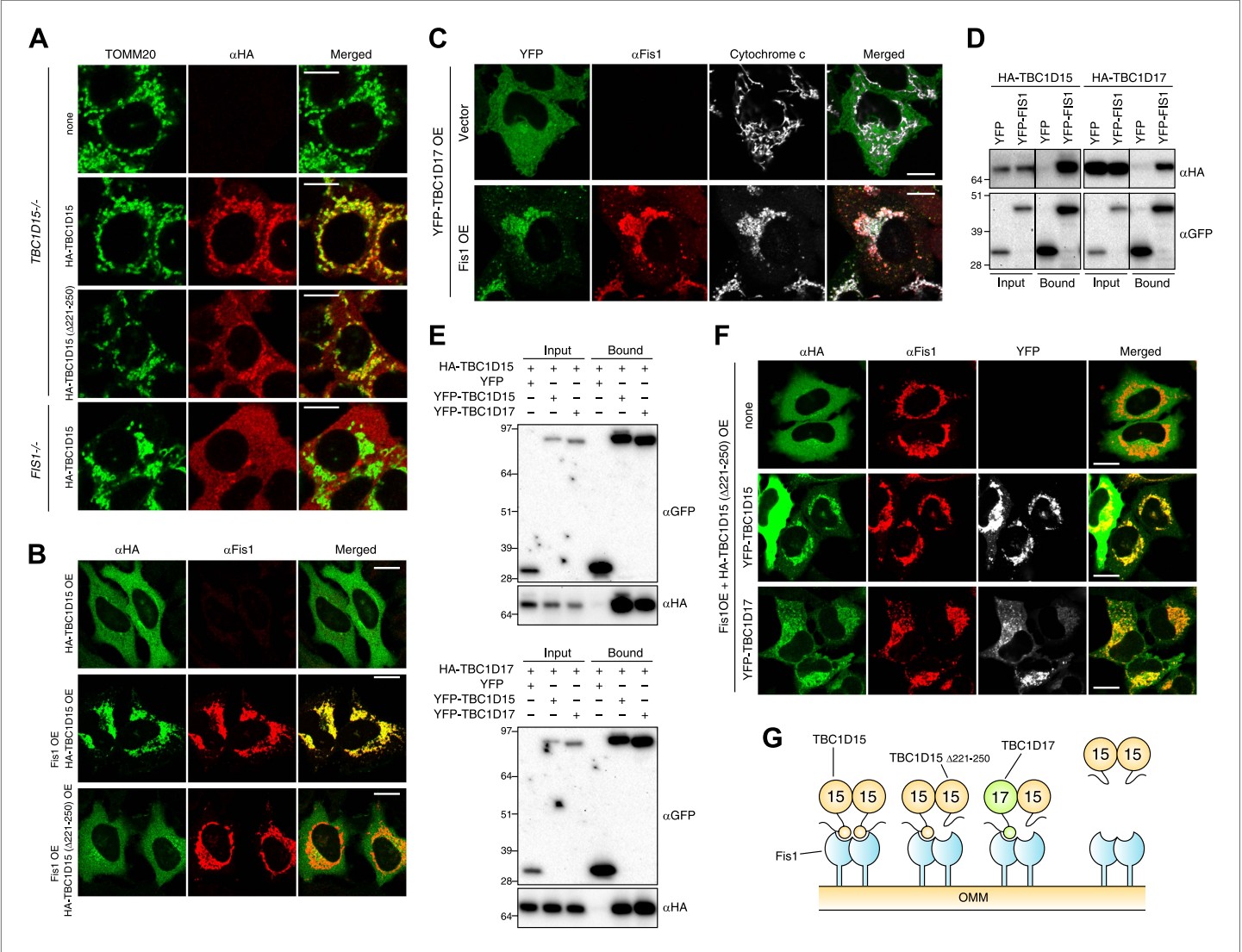

**Figure 6**. Identification of TBC1D17 as a Fis1 and TBC1D15 binding protein. (**A**) *TBC1D15−/−* cells and those stably expressing HA-TBC1D15 WT and HA-TBC1D15 (Δ221-250), and *FIS1−/−* cells stably expressing HA-TBC1D15 WT were subjected to immunostaining with anti-TOMM20 and anti-HA antibodies. Scale bars, 10 μm. (**B**) HA-TBC1D15 WT or HA-TBC1D15 (Δ221-250) with or without Fis1 was transiently overexpressed (OE) in HeLa cells. Cells were subjected to immunostaining with anti-HA and anti-Fis1 antibodies. Scale bars, 20 μm. (**C**) YFP-TBC1D17 together with pcDNA vector or Fis1 was transiently overexpressed (OE) in HeLa cells. Cells were subjected to immunostaining with anti-Fis1 and anti-Cytochrome c antibodies. Scale bars, 20 μm. (**D**) YFP or YFP-Fis1 was co-overexpressed with HA-TBC1D15 or HA-TBC1D17 in HEK293 cells. The cell extracts were subjected to pull down assays with GFP-Trap. 5% input and bound fractions were analyzed by immunoblotting with anti-HA (upper panel) and anti-GFP (lower panel) antibodies. (**E**) YFP, YFP-TBC1D15, or YFP-TBC1D17 was co-overexpressed with HA-TBC1D15 (upper panel) or HA-TBC1D17 (lower panel) in HEK293 cells. The cell extracts were subjected to pull down assays with GFP-Trap. 5% input and bound fractions were analyzed by immunoblotting with anti-GFP and anti-HA antibodies. (**F**) HA-TBC1D15 (Δ221-250) together with Fis1 and YFP-TBC1D15 WT or YFP-TBC1D17 WT were transiently overexpressed (OE) in HeLa cells. Cells were subjected to immunostaining with anti-HA and anti-Fis1 antibodies. Images of HA and Fis1 staining were merged in the right panels. Scale bars, 20 μm. (**G**) Schematic model of Fis1, TBC1D15, and TBC1D17 binding. Homo- or hetero-dimer of TBC1D15 can interact with Fis1 dimer on the mitochondrial outer membrane (OMM).

The following figure supplements are available for figure 6:

**Figure supplement 1**. TBC1D17 gene knock out by TALENs.

**Figure supplement 2**. TBC1D17 is dispensable for normal mitochondria and peroxisome morphologies.

**Figure supplement 3**. YFP-LC3 accumulation in *TBC1D17−/−* and *TBC1D15/17 DKO* cells during mitophagy.

HA-TBC1D15 Δ221-250 could rescue LC3 accumulation in *TBC1D15−/−* cells during mitophagy (*Figure 2D,E*). However, TBC1D15 appears to bind Fis1 as reported (*Onoue et al., 2013*) because in a *FIS1−/−* background, TBC1D15 localizes to the cytosol (*Figure 6A*). Furthermore, Fis1 overexpression can recruit overexpressed cytosolic WT TBC1D15, but not TBC1D15 Δ221-250, to the mitochondria (*Figure 6B*). These results raise the possibility that there is another protein that can facilitate TBC1D15 Δ221-250 binding to Fis1.

The human genome encodes at least 40 different TBC domain-containing proteins. According to phylogenetic and domain analysis of human and mouse TBC proteins (*Itoh et al., 2006*), TBC1D17 has a high similarity to the primary protein sequence of TBC1D15. Specifically, the region outside the TBC domain comprising 193-312 aa of TBC1D17 is 35% identical (55% positive) to the 201-333 aa region of TBC1D15 that contains the Fis1-interacting region (data not shown), suggesting that TBC1D17 may also interact with Fis1. To test this idea, we expressed HA-tagged TBC1D17 (HA-TBC1D17) with or without Fis1 overexpression in HeLa cells. Overexpressed HA-TBC1D17 alone was cytosolic (*Figure 6C*). However, after co-overexpression with Fis1, HA-TBC1D17 localized to mitochondria (*Figure 6C*), indicating that like TBC1D15, TBC1D17 has an ability to bind Fis1. Similar results were obtained by co-immunoprecipitation with YFP-Fis1 (*Figure 6D*). Although we were unable to detect endogenous TBC1D17 because no antibody is available, these results suggest that TBC1D17 may also be targeted to mitochondria via an interaction with Fis1. By co-immunoprecipitation, we found that TBC1D15 and TBC1D17 can form homo- and hetero-dimers (*Figure 6E*). Furthermore, the TBC1D15 Δ221-250 mutant that lacks a Fis1-binding region translocates to mitochondria when co-overexpressed with Fis1 and WT TBC1D15 or WT TBC1D17 (*Figure 6F*), indicating that TBC1D17 can recruit the mutant TBC1D15 to Fis1 through dimerization. These results are also interesting in light of evidence that Fis1 also forms dimers (*Jofuku et al., 2005*; *Lees et al., 2012*) suggesting a new model of Fis1/TBC topology (*Figure 6G*).

To examine the functional importance of TBC1D17 in mitophagy, we generated *TBC1D17* single KO and *TBC1D15/TBC1D17* double KO (DKO) HCT116 cells by the TALEN methodology (*Figure 6—figure supplement 1*). Mitochondrial and peroxisomal morphologies in these cells were indistinguishable from those in WT cells (*Figure 6—figure supplement 2A,B*). *TBC1D17−/−* cells displayed LC3 accumulation during mitophagy similar to, but less severe than that of *TBC1D15−/−* cells (*Figure 6—figure supplement 3A,B*). However, the degree of LC3 accumulation in *TBC1D15/TBC1D17* DKO cells was not further increased as compared to that of *TBC1D15* single KO. Therefore, in HCT116 cells, TBC1D15 appears to be dominant for contouring autophagosome morphogenesis. TBC1D15 and TBC1D17 may have different cell type expression profiles or different regulatory mechanisms. Overexpression of either TBC1D15 or TBC1D17 can rescue the LC3 accumulation in *TBC1D15/17DKO* cells, whereas the TBC1D15 Δ221-250 mutant cannot (*Figure 6—figure supplement 3C,D*). Thus, Fis1 binding is required for TBC1D15 to govern isolation membrane biogenesis during mitophagy.

## TBC1D15 binds Atg8 family proteins

The absence of TBC1D15 causes the accumulation and/or tubulation of the autophagosome marker, LC3 upon Parkin activation of mitophagy. This would be consistent with Rab-GAP activity of TBC1D15 switching a Rab protein to an inactive form to counterbalance Parkin-induced Rab activation. We hypothesized that the localization of such a Rab-GAP to mitochondria could inhibit Rab activity on one side of the isolation membrane to help contour the membrane expansion to surround the cargo. TBC1D15 may coordinate mitochondrial attachment to pre-autophagosomal membranes with asymmetric inhibition of Rab proteins to guide autophagosomes to tightly engulf mitochondria. Consistent with this hypothesis, several TBC family proteins such as TBC1D25/OATL1 and TBC1D2B have been reported to directly bind Atg8 family proteins (*Itoh et al., 2011*; *Popovic et al., 2012*) and TBC1D15 was identified by proteomics to bind Atg8 family members (*Behrends et al., 2010*). To test this model, we assessed Atg8 family protein interaction with TBC1D15 and sought to identify novel LC3-interacting region (LIR) motifs. In mammals, there are at least six different ATG8 homologues including LC3A, LC3B, LC3C, GABARAP, GABARAPL1 and GABARAPL2. Both LC3 and GABARAP subfamilies are found on autophagosomal membranes (*Kabeya et al., 2004*). Similar to YFP-LC3B (*Figure 2A*), YFP-GABARAPL1 also forms massively expanded structures in *FIS1−/−* and in *TBC1D15−/−* cells after Parkin translocation induced by valinomycin treatment (*Figure 7A*). We prepared recombinant GST-tagged Atg8 family proteins and subjected them to in vitro binding assays with cell extracts overexpressing

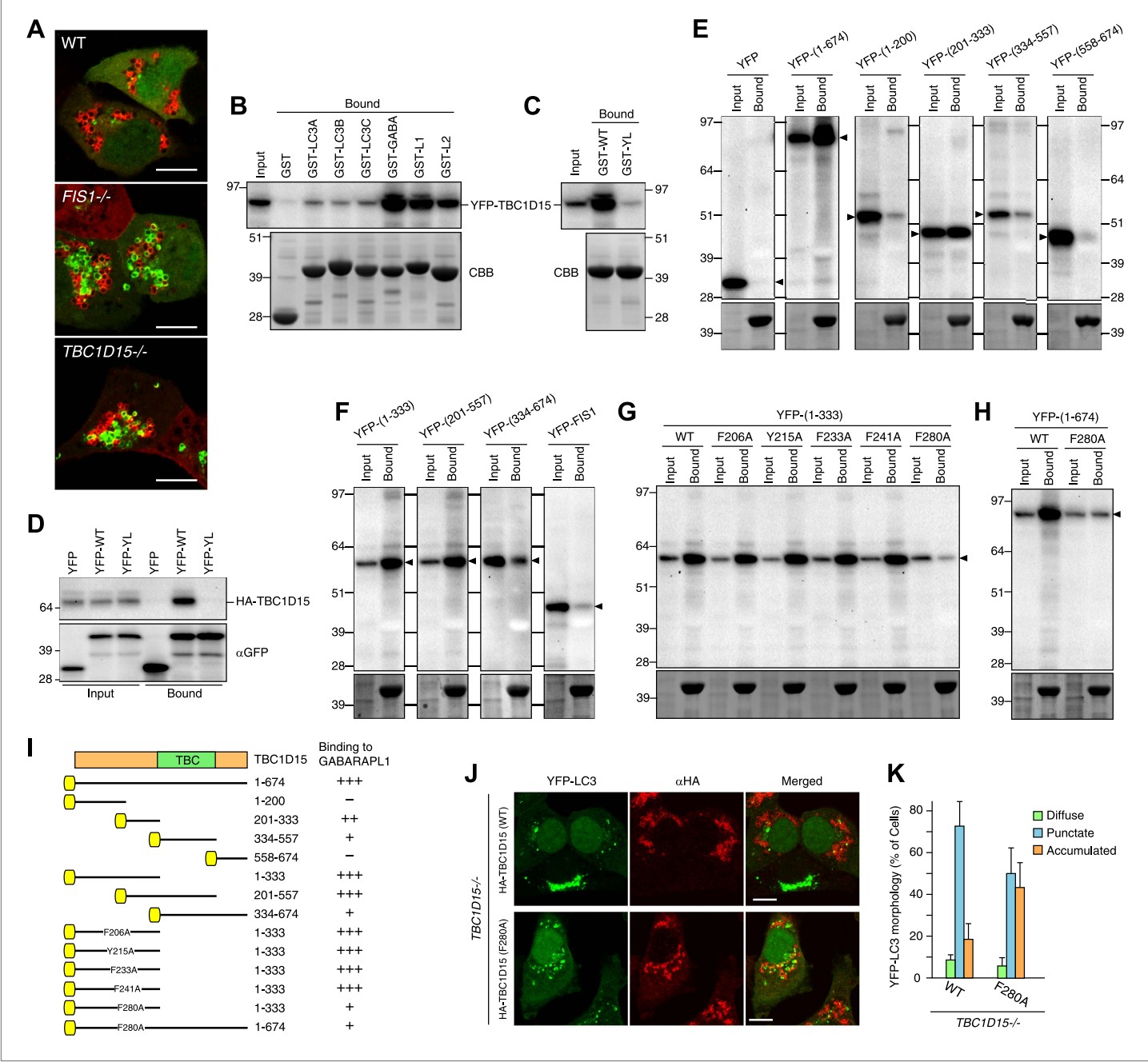

**Figure 7**. TBC1D15 binds ATG8 family proteins. (**A**) The indicated cells transiently expressing YFP-GABARAPL1 (green) and mCherry-Parkin (red) were treated with valinomycin for 3 hr. Scale bars, 10 µm. (**B**) YFP-TBC1D15 overexpressed in HEK293 cells was subjected to binding assays with GST-fused proteins (GABA, L1, and L2 represent GABARAP, GABARAPL1, and GABARAPL2, respectively). 5% input and bound fractions were analyzed by immunoblotting with anti-GFP antibody (upper panel). Coomassie brilliant blue (CBB) staining shows GST-fusion proteins in bound fractions (lower panel). (**C**) Binding assay carried out as in (**B**) with GST-GABARAPL1 WT (GST-WT) or its Y49A/L50A mutant (GST-YL). Immunoblotting with anti-GFP antibody (upper panel) and CBB staining (lower panel) are shown. (**D**) Cell extracts from HEK293 overexpressed HA-TBC1D15 and YFP, YFP-GABARAPL1 (YFP-WT), or its Y49AL50A mutant (YFP-YL) were subjected to pull down assays with GFP-Trap. 5% input and bound fractions were analyzed by immunoblotting with anti-HA (upper panel) and anti-GFP (lower panel) antibodies. (**E–H**) The indicated YFP-tagged TBC1D15 full-length, truncated, or point-mutant protein or YFP-Fis1 overexpressed in HEK293 cells were subjected to binding assays with recombinant GST-GABARAPL1. 5% input and bound fractions were analyzed by immunoblotting with anti-GFP antibody (upper panel) and CBB staining (lower panel). (**I**) Summary of binding abilities of truncated or point-mutated TBC1D15 constructs. −, +, ++, and +++ indicates binding of recombinant GST-GABARAPL1 to less than 1%, 1–5%, 5–10%, and over 10%, respectively of the total YFP-TBC1D15 fragment. Yellow boxes indicate YFP tags. (**J**) YFP-LC3 and mCherry-Parkin stably expressing *TBC1D15−/−* cells in
*Figure 7. Continued on next page*

*Figure 7. Continued*

the presence of HA-tagged TBC1D15 WT or F280A mutant were treated with valinomycin for 3 hr. Cells were subjected to immunofluorescence microscopy with anti-HA antibody. Scale bars, 10 µm. (**K**) The YFP-LC3 morphology of cells in (**J**) was quantified. The error bars represent ±SD from three independent replicates. Over 50 cells were counted in each well.

The following figure supplements are available for figure 7:

**Figure supplement 1**. Schematic representation of potential LIR motifs of TBC1D15.

**Figure supplement 2**. TBC1D17 has the ability to bind Atg8 homologues.

YFP-TBC1D15. While GST alone does not bind YFP-TBC1D15, LC3, and GABARAP subfamily members bind in moderate and high amounts to YFP-TBC1D15, respectively (*Figure 7B*). Fis1 cannot bind GABARAPL1, confirming that the binding between TBC1D15 and GABARAPL1 is not mediated indirectly through Fis1 (*Figure 7F*). In general, Atg8 homologues bind through their LDS (LIR docking site) to an LIR motif of a substrate (*Noda et al., 2008*). We performed more detailed analyses of the mechanism of GABARAPL1 binding to TBC1D15 because the GABARAP subfamily proteins display greater binding than LC3 subfamily members (*Figure 7B*). We made a GABARAPL1 LDS mutant by replacing Y49 and L50 with alanines. Although equal amounts of GST-tagged GABARAPL1 WT and LDS mutant were expressed and immunoprecipitated, the LDS mutant failed to bind YFP-TBC1D15 (*Figure 7C*), indicating that the binding occurs through LIR–LDS interactions. This result was confirmed in binding assays of HA-TBC1D15 and YFP-GABARAPL1 expressed in cultured cells (*Figure 7D*).

To explore the LDS–LIR interactions in more detail, we assessed potential LIR motifs in TBC1D15. The core of the LIR motif consists of the sequence (W/F/Y)XX(L/I/V), where X is any amino acid and with acidic amino acid residues often found near the LIR motif (*Alemu et al., 2012*). Based on this consensus motif, we identified 20 different LIR candidates in the TBC1D15 protein sequence (*Figure 7— figure supplement 1*). To narrow down candidates, we made and tested TBC1D15 truncations by in vitro binding assays using GST-tagged GABARAPL1. TBC1D15 was first divided into four parts according to the domain structure; the N-terminus (1–200), a region including the Fis1-interacting site (201-333), the TBC domain (334–557), and a C-terminal region (558–674), all of which were tagged with YFP at the N-terminus for detection by immunoblotting. Transiently overexpressed TBC1D15 domains in HEK293 cells subjected to in vitro binding assays revealed that only the 201–333 region retained the affinity for GABARAPL1 (*Figure 7E*). Similar results were obtained when we made longer constructs containing this region also indicating that polypeptides within the 201–333 aa region could efficiently bind GABARAPL1 (*Figure 7F*). Five different LIR candidates were found in the 201–333 aa region. By mutating the aromatic amino acid in each of these five LIR candidates to alanine, we identified $^{280}$FEVI$^{283}$ of TBC1D15 as the LIR motif responsible for full length TBC1D15 binding to GABARAPL1 (*Figure 7G,H*). A summary of TBC1D15 binding to GABARAPL1 is shown in *Figure 7I*.

To examine whether Atg8 binding to TBC1D15 is required to orchestrate autophagosome formation, the TBC1D15 F280A mutant was introduced into *TBC1D15−/−* cells. The F280A mutation resides outside the Fis1-binding region (221–250 aa of TBC1D15) (*Onoue et al., 2013*) and does not affect the mitochondrial targeting of TBC1D15 (data not shown). The LC3 membrane accumulation phenotype of the *TBC1D15−/−* cells was only partially rescued by the TBC1D15 F280A mutant (*Figure 7J,K*) relative to the complete rescue seen with WT TBC1D15, indicating that the interaction between TBC1D15 and Atg8 family proteins is important for autophagosome morphogenesis during Parkin-mediated mitophagy.

Interestingly, amino acids 261–264 of TBC1D17 are identical to the sequence of the LIR motif in TBC1D15 (FEVI) (*Figure 7—figure supplement 2A*). To test whether TBC1D17 also has the ability to bind Atg8 family proteins, in vitro binding assays were performed. Both LC3 and GABARAP subfamilies pulled down YFP-TBC1D17 (*Figure 7—figure supplement 2B*), and this interaction was mediated by the LDS domain of ATG8 family proteins (*Figure 7—figure supplement 2C,D*) with less discrimination between GABARAP and LC3 subfamily members than TBC1D15 (*Figure 7B*). Furthermore, the LIR mutation F261A prevented binding of TBC1D17 to GST-LC3B (*Figure 7—figure supplement 2E*).

These results indicate that both TBC1D15 and TBC1D17 have the ability to bind ATG8 family proteins via a LIR–LDS interaction.

## Rab7 is involved in autophagosomal membrane formation during mitophagy

As Rab-GAPs, TBC family proteins play important roles by inactivating Rab proteins and different Rab proteins are thought to be regulated by different TBC family members. Although little is known about the pairing between various Rab proteins and TBC proteins, in vitro GTPase assays (*Zhang et al., 2005*) and pull-down assays in cells (*Peralta et al., 2010*) suggest that TBC1D15 may function as a Rab-GAP for Rab7. Rab7 is required for endosome maturation and transport from the late endosome to the lysosome (*Bento et al., 2013*). However, Rab7 also has been reported to be involved in the biogenesis of autophagosomes (*Gutierrez et al., 2004*; *Jager et al., 2004*). We hypothesized that if TBC1D15 bound to Fis1 on mitochondria normally inhibits Rab7 during mitophagy through Rab-GAP activity, the excessive LC3-labeled tubulation observed in *TBC1D15−/−* cells (as well as *FIS1−/−* cells) might be a consequence of excessive or unregulated Rab7 activity. To test this hypothesis, we knocked down Rab7 by siRNA. We found 2 of 4 different siRNAs tested could knock down endogenous Rab7 expression efficiently (*Figure 8A*). These two Rab7 siRNAs were able to completely suppress the abnormal LC3 accumulation and tubulation in both *FIS1−/−* or *TBC1D15−/−* cells (*Figure 8B,C*, *Figure 8—figure supplement 1A* and data not shown). We investigated Rab7 localization in WT and *TBC1D15−/−* cells with or without valinomycin treatment. As we could not detect endogenous Rab7 by immunofluoresence microscopy, we utilized N-terminally 2HA- or YFP-tagged Rab7. In WT or *TBC1D15−/−* cells stably expressing YFP-LC3, mCherry-Parkin and 2HA-Rab7, Rab7 predominantly localizes on lysosomes with a weak ER signal both in WT and *TBC1D15−/−* cells under basal conditions (*Figure 8—figure supplement 1B*). However, after 3 hr valinomycin treatment of WT cells, 2HA-Rab7 colocalized with some of the YFP-LC3 in WT cells in dot or crescent-shaped structures that represent isolation membranes, indicating that Rab7 is targeted to isolation membranes upon stimulation of mitophagy (*Figure 8D*, WT valinomycin). Interestingly, Rab7 was also found on YFP-LC3 positive tubules in *TBC1D15−/−* cells (*Figure 8D*, *TBC1D15−/−* valinomycin), strongly supporting the hypothesis that Rab7 functions in isolation membrane expansion during mitophagy.

We next asked whether the activity of TBC1D15 was dependent on Rab-GAP activity. The mutant HA-tagged TBC1D15 D397A, which lacks TBC domain GAP activity (*Rak et al., 2000*), was compared with WT TBC1D15 for the ability to rescue *TBC1D15−/−* cells from excessive LC3 accumulation during Parkin-mediated mitophagy. Although HA-TBC1D15 D397A localizes on mitochondria (data not shown), it fails to rescue the accumulation of LC3 in *TBC1D15−/−* cells as WT TBC1D15 does (*Figure 8E,F*). Furthermore, overexpression of TBC1D17 catalytic mutant R381A could not rescue the LC3 accumulation in the *TBC1D15/TBC1D17* DKO cells (*Figure 8—figure supplement 2A,B*). Although many different Rab proteins such as Rab5, Rab8, Rab21 and Rab35 have been suggested to be the targets of TBC1D17 (*Itoh et al., 2006*; *Fuchs et al., 2007*; *Vaibhava et al., 2012*), Rab7 siRNA could suppress YFP-LC3 accumulation in *TBC1D17−/−* cells (*Figure 8—figure supplement 2C,D*). These results support the model that Rab-GAP activity inhibits Rab7 to mitigate excessive autophagosome membrane expansion during mitophagy and, through Fis1 binding to mitochondria and LC3 family member binding on isolation membranes, tailors the autophagosomal membrane expansion to its cargo.

## Discussion

In this study, we identify Rab7, TBC1D15 and TBC1D17, and Fis1 as mitophagy effectors functioning downstream of Parkin to regulate autophagosome morphogenesis. Mitochondrial damage is initially signaled to Parkin through the kinase PINK1 (*Youle and Narendra, 2011*). As PINK1 accumulates on the outer membrane of depolarized mitochondria, it recruits Parkin and activates Parkin ubiquitin ligase activity (*Iguchi et al., 2013*; *Lazarou et al., 2013*; *Zheng and Hunter, 2013*). Parkin then modifies numerous mitochondrial outer membrane proteins with K48- and K63-linked ubiquitin chains. This ubiquitin signal appears to recruit ULK1, Atg9 and LC3 to the mitochondrion independently of one another to induce mitophagy (*Itakura et al., 2012*). Our results also indicate that Rab7 is likely activated by a Rab GEF to promote autophagosomal membrane growth and microtubule associated trafficking and that TBC1D15/17 Rab GAP activity is required to temper

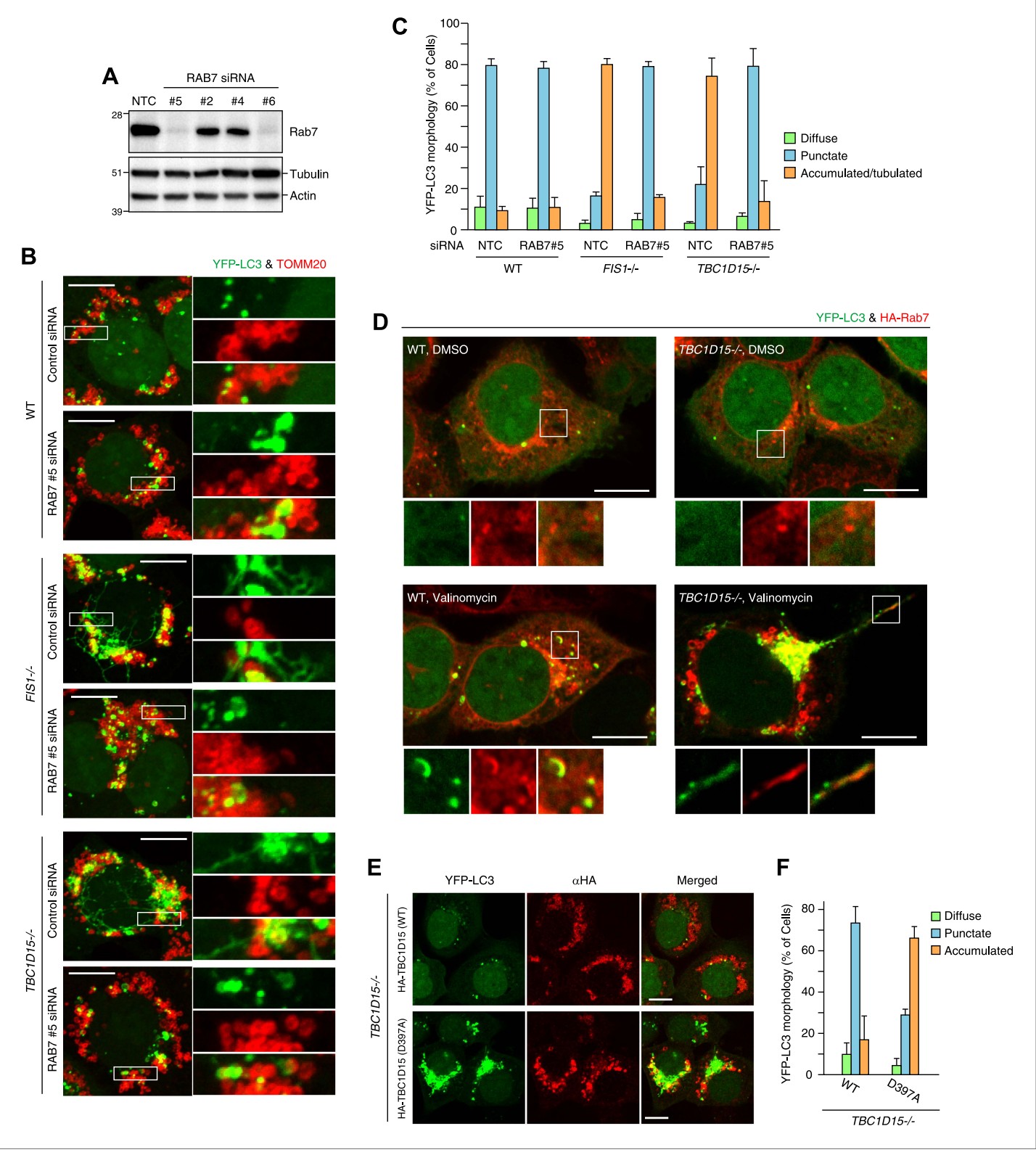

**Figure 8**. Rab7 is involved in autophagosome fusion during mitophagy. (**A**) Total cell lysates from HCT116 treated with control (NTC) or RAB7A siRNA were analyzed by immunoblotting. (**B**) The indicated cells stably expressing YFP-LC3 (green) and mCherry-Parkin were treated with control (NTC) or Rab7_#5 siRNA. After 3 hr valinomycin treatment, cells were analyzed by immunofluorescence microscopy with anti-TOMM20 antibody (red). Z-stacks of confocal images are shown. Magnified images are also shown. Scale bars, 10 μm. (**C**) YFP-LC3 morphologies of cells in (**B**) were quantified. Percentages

*Figure 8. Continued on next page*

*Figure 8. Continued*

of cells harboring diffuse, punctuate or accumulated/tubulated YFP-LC3 are shown. Data and error bars were obtained from at least 50 cells in each of three independent replicates. (**D**) The indicated cells stably expressing YFP-LC3 (green), mCherry-Parkin, and 2HA-Rab7 (Red) were treated with or without valinomycin for 3 hr and analyzed by immunofluorescence microscopy with anti-HA antibody. Magnified images are also shown. Scale bars, 10 µm. (**E**) YFP-LC3 and mCherry-Parkin stably expressing *TBC1D15*−/− cells in the presence of HA-tagged TBC1D15 WT or the D397A mutant were treated with valinomycin for 3 hr. Cells were subjected to immunofluorescence microscopy with anti-HA antibody. Scale bars, 10 µm. (**F**) The YFP-LC3 morphology of cells in (**E**) was quantified. The error bars represent ±SD from three independent replicates. Over 50 cells were counted in each replicate.

The following figure supplements are available for figure 8:

**Figure supplement 1**. Rab7 is involved in LC3 accumulation of *FIS1*−/− and *TBC1D15*−/− cells.

**Figure supplement 2**. TBC1D17 GAP activity is important for LC3 accumulation through Rab7.

Rab7 activity to tailor autophagosomal membrane expansion to surround mitochondrial aggregates. Without TBC1D15, or with TBC1D15 lacking Rab GAP activity, LC3/GABARAP-labeled isolation membranes accumulate excessively and without cargo orientation sending long tubules away from mitochondria along microtubule tracks. This suggests that Rab GAP activity localized on the cargo—facing only one side of the expanding LC3 positive isolation membranes—may inhibit Rab7 activity and thereby inhibit membrane expansion proximal to the cargo forcing the growing membrane to tightly wrap odd shaped cargo. Alternatively, perhaps LC3-decorated preautophagosomal structures traffic along microtubules toward Parkin-ubiquitinated mitochondria and fuse together around mitochondrial aggregates where TBC1D15 bound to mitochondria inhibits Rab7 to mediate the release of LC3-bound membranes from microtubules as they contact the mitochondrial cargo (**Pankiv et al., 2010**).

*TBC1D15*−/− cells display normal LC3 accumulation during starvation suggesting that these membrane expansion steps are specific for mitophagy or other forms of selective autophagy. Xenophagy, for example, also requires Rab proteins for sequestering invading bacteria in autophagosome-like vacuoles (**Yamaguchi et al., 2009**), consistent with the idea that isolation membrane expansion is mediated by Rab activity. Parkin also may activate Rab proteins as a step in xenophagy (**Manzanillo et al., 2013**).

We identify an LIR domain in TBC1D15, conserved in TBC1D17, that is also required to restrain isolation membrane biogenesis to the surface of the mitochondria. Proteins that contain both an LIR domain and a ubiquitin-associated domain such as p62 (**Bjorkoy et al., 2005**), optineurin (**Wild et al., 2011**), NBR1 (**Kirkin et al., 2009**), and NDP52 (**Thurston et al., 2009**) can bind to LC3 and to polyubiquitin chains to recruit ubiquitinated cargo into nascent isolation membranes. TBC1D15 and TBC1D17 are also LIR domain-containing LC3/GABARAP-binding proteins. However, in contrast to the ubiquitin-binding LIR domain proteins, TBC1D15 and TBC1D17 appear to constitutively bind to mitochondria through Fis1 and thus, unless somehow regulated, would not appear to initiate the recruitment of LC3 to mitochondria. More likely, TBC1D15 and TBC1D17 use LC3/GABARAP binding in the context of Rab GAP activity to orient growing isolation membranes to the surface of the cargo. Interestingly, TBC1D15 and TBC1D17 appear to heterodimerize and homodimerize, which may increase their affinity for LC3/GABARAP by multivalent binding. Fis1 also forms dimers consistent with its role as the TBC1D15/17 receptor. The self association of TBC1D15 and TBC1D17 is shared by other ubiquitin-binding LIR domain proteins such as p62 (**Birgisdottir et al., 2013**).

We previously reported that *FIS1*−/− cells display no detectable mitochondrial morphology phenotype (**Otera et al., 2010**). However, it is well established that yeast Fis1 binds Dnm1 through Mdv1/Caf4 and regulates mitochondrial fission (**Okamoto and Shaw, 2005**). Furthermore, Fis1 loss in *C. elegans* causes LC3 accumulation and has been linked to stress-induced mitochondrial fission (**Shen et al., 2014**). Interestingly, Dnm1 activity in yeast (**Mao et al., 2013**), and Drp1 activity in mammalian cells is required for mitophagy (**Twig et al., 2008**; **Tanaka et al., 2010**; **Frank et al., 2012**) suggesting a link with Fis1 in mitophagy. In contrast to human Fis1, yeast Fis1 has an N-terminal extension that folds back into a pocket used by other tetratricopeptide motif proteins to bind ligands (**Suzuki et al., 2003**, **2005**). It will be important to determine how TBC1D15 interacts with human Fis1 and if the Rab GAP-binding activity of human Fis1 is conserved in yeast. Fis1 also localizes to peroxisomes and would

be predicted to recruit TBC1D15/17 to peroxisomes as well, suggesting that Fis1 and TBC1D15/17 may also play a role in pexophagy.

## Materials and methods

### Plasmids

Human *LC3B* was cloned into BglII/EcoRI sites of pEYFP-C1 (Clontech, Mountain View, CA) to make YFP-LC3, which was then cloned into BamHI/NotI sites of pCHAC/IRES (Allele Biotechnology, San Diego, CA) to make a retrovirus plasmid pCHAC/YFP-LC3B-IRES-MCS2. To generate a retrovirus mCherry-Parkin plasmid (pBMNz/mCherry-Parkin), the gene for N-terminally mCherry-tagged Parkin (*Narendra et al., 2008*) was cloned into BamHI/NotI sites of pBMN-Z vector (Addgene plasmid 1734). 2×HA-tagged RAB7A gene from DsRed-rab7 WT (Addgene plasmid 12661) was subcloned into pBABE-puro vector (Addgene plasmid 1764) using BamHI to generate pBABE-puro/2HA-Rab7. YFP-GABARAPL1 was generated by inserting a *GABARAPL1* gene into BglII/EcoRI sites of pEYFP-C1 vector. The Y49AL50A mutation in *GABARAPL1* and the F52AL53A mutation in *LC3B* were introduced by primer-based PCR mutagenesis. For GST expression, pGEX-KG vector was used. The following plasmids were also used in this study: pMRX-IP GFP-ULK1 and pMRXs-puro GFP-DFCP1 (kind gifts from Dr Noboru Mizushima), pMXs-IP GFP-Atg14 (Addgene plasmid 38264)

To generate YFP-TBC1D15 and YFP-TBC1D17 plasmids, *TBC1D15* and *TBC1D17* genes were amplified from HA-TBC1D15 plasmid (a kind gift from Dr Naotada Ishihara) and from POTB7/TBC1D17 (purchased from Thermo Scientific #MHS6278-202827046) by PCR and inserted into the XhoI/KpnI sites of pEYFP-C1 plasmid. YFP-TBC1D15-truncated mutants were generated by the same method with appropriate primer pairs. Alanine mutants were generated by primer-based PCR mutagenesis. Construction of HA-TBC1D15 WT and HA-TBC1D17 was done as described in *Onoue et al. (2013)*. R381A mutation in HA-TBC1D17, F280A or D397A mutations in HA-TBC1D15 and deletion of 221–250 aa in HA-TBC1D15 were achieved by primer-based PCR mutagenesis. These *HA-TBC1D15* genes were then subcloned into the BamHI/EcoRI sites of pBMN-z. 3xFLAG-Fis1 was generated by cloning *3×FLAG* and *FIS1* genes into pBABE-puro.

### Antibodies

The following antibodies were used for immunoblotting: rabbit anti-GFP (A-11122; Invitrogen, Grand Island, NY), mouse anti-HA (clone 16B12; Covance, Berkeley, CA), mouse anti-α-Tubulin (Invitrogen clone B-5-1-2), mouse anti-Actin (clone AC-40; Sigma, St. Louis, MO), rabbit anti-TOMM20 (sc-11415; Santa Cruz Biotechnology, Inc., Dallas, TX), rabbit anti-PINK1 (BC100-494; Novus Biologicals, Littleton, CO), mouse anti-TIMM23 (clone 32; BD Biosciences, San Jose, CA), rabbit anti-LC3B (L7543; Sigma), mouse anti-HSP60 (clone LK-1; Stressgen, Victoria, Canada), rabbit anti-Mfn1 (generated as described previously in the study by *Karbowski et al., 2007*), mouse anti-Rab7 (Rab7-117; Abcam, Cambridge, MA), rabbit anti-TBC1D15 (described previously in the study by *Onoue et al., 2013*), rabbit anti-Fis1 (ALEXIS Biochemicals, San Diego, CA), rabbit anti-Mff (described previously in the study by *Gandre-Babbe and van der Bliek, 2008*), mouse anti-Drp1 (clone 8/DLP1; BD Biosciences), mouse anti-MTCOXII (clone 12C4F12; Abcam), and mouse anti-HA (clone 16B12; COVANCE). The following antibodies were used for immunostaining: rabbit anti-TOMM20 (sc-11415; Santa Cruz Biotechnology, Inc.), mouse anti-HA (clone 16B12; COVANCE), mouse anti-FLAG M2 (Agilent Technologies, Cedar Creek, TX), rabbit anti-Fis1 (ALEXIS Biochemicals), mouse anti-Cytochrome c (clone 6H2.B4; BD Biosciences), rabbit anti-PMP70 (71-8300; Invitrogen), rabbit anti-GFP antibody (A-11122; Invitrogen), mouse anti-α-Tubulin (Invitrogen clone B-5-1-2), mouse anti-LAMP2 antibody (sc-18822; Santa Cruz Biotechnology, Inc.), mouse anti-GM130 antibody (clone 35/GM130; BD Biosciences), mouse anti-KDEL antibody (clone 10C3; Stressgen), and rabbit anti-Atg16L1 antibody (a kind gift from Dr Noboru Mizushima). F-Actin was stained with Alexa Fluor 647 Phalloidin (invitrogen).

### RNA interference

RAB7A siRNA oligos were purchased from QIAGEN (Valencia, CA). The target sequences are as follows: RAB7_#5, CACGTAGGCCTTCAACACAAT, RAB7_#6, CTGCTGCGTTCTGGTATTTGA, RAB7_#2, TCCCGTTAGATCAGCATTCTA, RAB7_#4, TAGATCAGCATTCTACTACAA. Nontargeting control siRNA and PINK1 siRNA were described previously (*Lazarou et al., 2013*).

## Cell culture and transfection

HeLa, HEK293, and HEK293T cells were cultured in Dulbecco's modified Eagle's medium (DMEM) supplemented with 10% (vol/vol) fetal calf serum, 10 mM HEPES buffer, 1 mM sodium pyruvate, 1 mM glutamine, and nonessential amino acids. HCT116 cells were cultured in McCoy's 5A medium supplemented with 10% (vol/vol) fetal calf serum, 1 mM glutamine, and nonessential amino acids. The cells were cultured at 37°C in a 5% $CO_2$ incubator. *FIS1−/−* HCT116 cell was reported previously (*Otera et al., 2010*). For transient transfection of plasmids, Fugene HD (Promega, Madison, WI) transfection reagent was used according to the manufacture's instruction. For gene knock-down by siRNA, the cells were transfected with nontargeting control or *RAB7A* siRNA (at a final concentration of 12.5 nM) or *PINK1* siRNA (at a final concentration of 6 nM) with Lipofectamine RNAiMAX (Invitrogen) according to the manufacture's instruction. After 24 hr, the medium was changed to fresh medium and the cells were grown for further 24 hr before analysis. Stable cell lines were established by recombinant retrovirus infection as follows. Vector particles were produced in HEK293T cells grown in poly-lysine-coated plates by cotransfection with Gag-Pol, VSV-G and a retrovirus plasmid (pCHAC/YFP-LC3B-IRES-MCS2, pBMNz/mCherry-Parkin, pBABE-puro/2HA-Rab7, pBABE-puro/YFP-Rab7, pBABE-puro/3FLAG-Fis1, pMRX-IP GFP-ULK1, pMRXs-puro GFP-DFCP1, pMXs-IP GFP-Atg14, pBMN-z/HA-TBC1D15 or its mutants). After 12 hr of transfection, the medium was changed to a fresh medium and the cells were further incubated for 24 hr. The viral supernatants were then infected into HCT116 cells with 8 µg/ml polybrene (Sigma).

Amino acid starvation was induced by washing cells twice with starvation buffer (20 mM HEPES pH 7.4, 140 mM NaCl, 1 mM $CaCl_2$, 1 mM $MgCl_2$, 5 mM glucose) followed by incubation with starvation buffer containing 1% (wt/vol) BSA.

Valinomycin (Sigma) was used at a final concentration of 10 µM. When cells were treated with valinomycin more than 6 hr, 10 µM Q-VD-OPH (SM Biochemicals, Anaheim, CA) was added to block apoptopic cell death. Nocodazole (Calbiochem, Darmstadt, Germany) was used at a final concentration of 10 µM.

## Construction of knock out cell line by TALENs

The *MFF−/−* cell line constructed by TALENs is described in *Shen et al., 2014* and the *DRP1−/−* cell line constructed by TALENs is described in Wang et al. (unpublished). The *TBC1D15−/−* cell line was generated by TALEN targeting the following site: 5'- GCC CTG TTG TTC AAA GGA GAG Aac cgg tAT CAC TGG AAG AAT GGA CTA AGA ACA TTG -3' (underlined sequences and lower-case sequences indicate the target sites for left and right TALEs and AgeI restriction enzyme site, respectively). The targeting site for knock out of *TBC1D17* gene was 5'- CTT CCC CCA ACA CAG TGC CGT CTC cct agg TGC AGA GCC CAG CTG CCC CCA GG -3' (AvrII localized at the junction between intron and exon5 was used for restriction enzyme site). 16-mer left and right TALEs were assembled according to *Huang et al. (2011)* and cloned into a final TALEN vector modified from *Miller et al. (2011)*. 0.8 µg of left and right TALEN constructs were cotransfected with 0.4 µg pEYFP-C1 vector in HCT116 cells and the cells were grown for 2 days. YFP-positive cells were sorted by FACS and plated into 96-well plates. Genomic DNA was isolated from single colonies, PCR amplification of the target site was performed, and assayed by restriction enzyme digestion. Finally, the gene knock out clones were confirmed by DNA sequencing and/or immunoblotting.

## Immunocytochemistry and confocal imaging

Cells grown on 2-well coverglass chamber slides were fixed with 4% paraformaldehyde in PBS for 25 min at room temperature, permeabilized with 0.015% (vol/vol) TX-100 in PBS for 15 min, and preincubated with 5% (wt/vol) BSA for 30 min. The fixed cells were incubated with primary antibodies and appropriate secondary antibodies (Alexa Fluor 488, 594 or 647 goat anti-rabbit or anti-mouse IgG from Invitrogen) for immunostaining. The images of the cells were captured using an inverted confocal microscope (LSM510 Meta, Carl Zeiss) with a 63×/1.4 NA oil differential interference contrast Plan-Apochromat objective lens. For image analysis, Volocity (PerkinElmer) and/or Photoshop (Adobe) software were used.

## Immunoblotting

For preparation of total cell lysate, cells grown in a six-well plate were washed twice with PBS and solubilized with 2% CHAPS buffer (25 mM HEPES-KOH pH 7.5, 300 mM NaCl, 2% (wt/vol) CHAPS,

protease inhibitor cocktail (Roche, Indianapolis, IN) on ice for 30 min and then protein concentrations were determined. Proteins precipitated with trichroloacetic acid were lysed with NuPAGE LDS sample buffer (Invitrogen) supplemented with 80 mM dithiothreitol. The appropriate amounts of proteins were applied and separated on 4–12% Bis-Tris SDS-PAGE (Invitrogen) with MES or MOPS SDS running buffer (Invitrogen). After transfer to PVDF membrane, blocking and incubation with primary antibodies, proteins were detected using horseradish peroxidase-coupled secondary antibodies (GE Healthcare Life Sciences, Piscataway, NJ) and ECL Plus or ECL Prime western blotting detection reagents (GE Healthcare Life Sciences).

### In vitro binding assay

GST alone or GST-tagged Atg8 family proteins were overexpressed in *Escherichia coli* strain BL21(DE3) from the plasmids, which were a gift from Dr Terje Johansen (*Pankiv et al., 2007*). Transformants were grown in LB medium supplemented with 100 mg/L ampicillin at 37°C and the protein expression was induced by the addition of 1 mM isopropyl β-$_D$-thiogalactopyranoside for 24 hr at 16°C. Cells were harvested by centrifugation, washed with PBS, and kept at −80°C before use. The cell pellets were lysed with B-PER bacterial extraction reagent (Pierce, Rockford, IL) supplemented with 100 μg/ml lysozyme and 5 units/ml DNaseI and incubated at room temperature for 10 min. After sonication, the cell debris and insoluble fraction were removed by centrifugation (20,000×$g$, 15 min, 4°C). The resulting supernatant was incubated with Glutathione Sepharose 4 Fast Flow (GE Healthcare Life Sciences) equilibrated with PBS for 1 hr at 4°C. GST-bound sepharose was washed three times with PBS-containing 500 mM NaCl. YFP-tagged TBC1D15 and its derivatives were overexpressed in HEK293 cells grown in a six-well plate. The cells were washed with PBS twice, and solubilized with 0.5% TX-100 buffer (50 mM HEPES pH 7.5, 150 mM NaCl, 1 mM EDTA, 1 mM EGTA, 8% [vol/vol] glycerol, 25 mM NaF, 0.5 [vol/vol] Triton X-100, protease inhibitor cocktail [Roche]) for 20 min on ice. Insoluble proteins were removed by centrifugation (20,000×$g$, 15 min, 4°C). The resulting supernatants were mixed with GST protein-bound sepharose in TBC-binding buffer (50 mM HEPES pH 7.5, 150 mM NaCl, 1 mM EDTA, 1 mM EGTA, 8% [vol/vol] glycerol, 25 mM NaF) for 3 hr at 4°C. The sepharose slurry was washed with 0.1% TX-100 buffer three times and proteins were eluted with NuPAGE LDS sample buffer (Invitrogen) containing 80 mM dithiothreitol.

The in vitro binding assay using GFP-Trap was performed as follows. YFP-tagged proteins and inter-acting candidates were co-overexpressed in HEK293 cells grown in six-well plates by transient transfection with Lipofectamine LTX (Invitrogen). After 24 hr of transfection, the cells were washed with PBS twice and solubilized with 0.5% TX-100 buffer for 20 min on ice. The resulting lysates were clarified by centrifugation and mixed with equilibrated GFP-Trap resin (Chromotek, Planegg-Martinsried, Germany) in TBC-binding buffer for 1 hr at 4°C. Proteins were washed with 0.1% TX-100 buffer three times before elution with LDS sample buffer containing 80 mM dithiothreitol.

### Immunoelectron microscopy

HCT116 WT and *TBC1D15*−/− cells stably expressing YFP-LC3 and mCherry-Parkin were treated with Valinomycin for 3 hr and fixed for 30 min with 4% paraformaldehyde and 0.1% glutaraldehyde in PBS. The fixed cells were washed four times with PBS, followed by permeabilization for 40 min with 0.1% Saponin and 5% goat serum in PBS. The cells were then incubated for 1 hr with mouse anti-GFP antibody (Invitrogen clone 3E6), followed by 1 hr with nanogold-conjugated anti-mouse IgG antibody (Nanoprobes) and further processing as described (*Tanner et al., 1996*). Thin sections (~80 nm) were counter stained with uranyl acetate and lead citrate. The sections were examined with a JEOL 200 CX transmission electron microscope. Images were collected with a digital CCD camera (AMT XR-100; Danvers, MA).

## Acknowledgements

We thank Dr Naotada Ishihara for TBC1D15 cDNA (HA-TBC1D15) and rabbit anti-TBC1D15 antibodies, Dr Terje Johansen for Atg8 homologue plasmids (pDEST15-LC3A, pDEST15-LC3B, pGEX4T-1-LC3C(DG), pDEST15-GABARAP, pDEST15-GABARAPL1, and pDEST15-GABARAPL2), Dr Noboru Mizushima for pMRX-IP GFP-ULK1, pMRXs-puro GFP-DFCP1, and rabbit Atg16L1 antibody, Dr Dragan Maric for cell sorting, Dr Michael Lazarou and members of the Youle laboratory for valuable discussions and comments. Technical support and advise for electron microscopy was provided by the NINDS EM facility.

# Additional information

## Competing interests

RJY: Reviewing editor, *eLife*. The other authors declare that no competing interests exist.

## Funding

| Funder | Grant reference number | Author |
|---|---|---|
| Japan Society for the Promotion of Science Postdoctoral Fellowship for Research Abroad | | Koji Yamano |
| National Institute of Neurological Disorders and Stroke intramural program | | Koji Yamano, Adam I Fogel, Chunxin Wang, Richard J Youle |
| National Institute of General Medical Sciences | GM051866 | Alexander M van der Bliek |

The funders had no role in study design, data collection and interpretation, or the decision to submit the work for publication.

## Author contributions

KY, Conception and design, Acquisition of data, Analysis and interpretation of data, Drafting or revising the article; AIF, Acquisition of data, Analysis and interpretation of data, Drafting or revising the article; CW, Acquisition of data, Drafting or revising the article; AMB, Analysis and interpretation of data, Drafting or revising the article, Contributed unpublished essential data or reagents; RJY, Conception and design, Analysis and interpretation of data, Drafting or revising the article

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
