## [Decision Letter]

Thank you for sending your work entitled “Mitochondrial Rab GAPs govern autophagosome biogenesis during mitophagy” for consideration at *eLife*. Your article has been favorably evaluated by a Senior editor and 3 reviewers, one of whom is a member of our Board of Reviewing Editors.

The Reviewing editor and the other reviewers discussed their comments before we reached this decision, and the Reviewing editor has assembled the following comments to help you prepare a revised submission.

The present manuscript by Yamano et al. reports interesting results to suggest a role for the mitochondrial Rab GAPs TBC1D15 and TBC1D17 in mitophagy. More specifically, TBC1D15 binds to Fis1 located in the outer mitochondrial membrane and to LC3/GABARAP protein(s) on the phagophore while restraining (inhibiting) Rab7 activity at the interface between mitochondria and the phagophore.

This report provides a novel mechanism how damaged mitochondria are recognized by autophagy. Overall, the manuscript is well written and the results are mostly convincing. However, there are some criticisms/concerns that need to be addressed before publication.

1) The authors use the term “expansion” in several ways, such as “isolation membrane expansion”, “autophagosomal expansion”, and “LC3 membrane expansion”. However, the nature of this process is unclear. It is possible that the expansion represent dynein-dependent transport of the abnormal mitochondria–autophagosome complex (LC3-positive mitochondria) or its clusters rather than elongation of the isolation membranes. It is even possible that these LC3-positive structures are aggregates/aggresomes of the LC3 protein, which can be formed in a microtubule-dependent manner. To address these issues, (1) the author should observe and compare isolation membrane marker proteins during mitophagy in control, *FIS1*^*-/-*^ cells, and *TBC1D15*^*-/-*^ cells, and (2) perform electron microcopy to clarify the nature of the expanded LC3 structures.

2) The effect of deletion of FIS1 or TBC1D15 on mitophagy is modest. The authors only show that the level of TOMM20 is higher in *FIS1*^*-/-*^ and *TBC1D15*^*-/-*^ cells compared to wild-type cells at 40 h after valinomycin treatment. As this is one of the main conclusions of this study, it should be more vigorously investigated, for example by quantification of mtDNA and other protein markers.

3) The role of Rab7 in mitophagy is also not convincing. Although there seems to be a significant difference in Figure 7, the images in Figure 7 do not convincingly support this quantification results. In particular, at which step mitophagosome formation is blocked is not clearly demonstrated. Again, electron microscopy can distinguish whether membrane formation or fusion is inhibited at an early or late step, or recognition of mitochondria by autophagic membranes is impaired.

4) The order of figures is not well structured and not easy to follow throughout the manuscript. For example, Figure 5 is referred prior to Figure 1, and parts of Figure 2 are discussed later in the manuscript after Figure 3. Please consider rearranging the figure to fit with the order of description in the text.

5) The authors need to describe how they define each group in Figure 1, Figure 2, Figure 2—figure supplement 2, Figure 2—figure supplement 3, Figure 3, Figure 4, Figure 5 and Figure 7. Especially, the criteria how the authors count diffuse, punctate, and aggregated LC3 should be clearly stated. The authors also need to state how many cells they counted for the statistical analyses.

---

## [Author Response]

*1) The authors use the term “expansion” in several ways, such as “isolation membrane expansion”, “autophagosomal expansion”, and “LC3 membrane expansion”. However, the nature of this process is unclear. It is possible that the expansion represent dynein-dependent transport of the abnormal mitochondria-autophagosome complex (LC3-positive mitochondria) or its clusters rather than elongation of the isolation membranes. It is even possible that these LC3-positive structures are aggregates/aggresomes of the LC3 protein, which can be formed in a microtubule-dependent manner. To address these issues, (1) the author should observe and compare isolation membrane marker proteins during mitophagy in control,* FIS1^−/−^
*cells, and* TBC1D15^−/−^
*cells, and (2) perform electron microcopy to clarify the nature of the expanded LC3 structures*.

We agree that we should use the term “expansion” more carefully. We also agree that the expansion may result from excessive dynein-dependent transport and have added more discussion of that to the manuscript. Based on our new electron microscopy data where we see LC3 associated with membranes and not in protein aggregates, we now consistently use the term “LC3 accumulation” when fluorescence microscopy and/or Western blotting shows higher levels of YFP-LC3. When we discuss the tubulation, we say “LC3-labeled tubules” instead of the term “membrane expansion” because we currently do not know the nature of the LC3-labeled tubules other than that they appear membranous (see new Figure 3B).

To address the nature of LC3 in more detail, we observed and compared isolation membrane marker proteins and upstream autophagy-related proteins by fluorescence microscopy as suggested. We made HCT116 WT, *FIS1*^*-/-*^ and *TBC1D15*^*-/-*^ cells stably expressing GFP-mouseULK1 (the most upstream ATG1/ULK kinase complex), GFP-mouseDFCP1 (PI3P-binding protein that can be used as an omegasome marker), and GFP-Atg14 (one of the subunits of Class III PI3K complex) and examined their accumulation during valinomycin-induced Parkin-mediated mitophagy. We also detected endogenous Atg16L1 that is associated with Atg12-Atg5 complex and serves as an isolation membrane marker. All of these labeled proteins were recruited adjacent to mitochondria by valinomycin treatment, but the number of dots or signals was similar among WT, *FIS1*^*-/-*^, and *TBC1D15*^*-/-*^ cells. These data are now presented as Figure 2—figure supplement 5 and Figure 2—figure supplement 6. These data suggest that in *FIS1*^*-/-*^ or in *TBC1D15*^*-/-*^ cells, only or primarily Atg8 family proteins accumulate during mitophagy. This is consistent with Itakura et al*.* (2012) J Cell Sci 125: 1488-99 showing that recruitment of upstream autophagy regulators such as ULK1, Atg14, DFCP1 and Atg16L1 and recruitment of LC3 are independent events during Parkin-mediated mitophagy and reveals that Fis1 and TBC1D15 function downstream of these steps. This is consistent with our model in which TBC1D15 and Rab7 function after Parkin translocation more proximal to isolation membrane expansion or trafficking.

To clarify the YFP-LC3 labeled structures both in WT and *TBC1D15*^*-/-*^ cells, we also performed immunoelectron microscopy. Valinomycin (3 hrs treatment) induced closely apposed preautophagosome membranes that are labeled with gold particles attached to YFP-LC3 in mCherry-Parkin stably expressing WT cells. *TBC1D15*^*-/-*^ cells also have similar cup-shaped preautophagosomal membranes, but many of them displayed much thicker lumens than those in WT cells. In addition, we saw more alternate types of LC3-labeled membranes in *TBC1D15*^*-/-*^ cells. Frequently, larger LC3-labeled membrane capsules were found that contained mitochondria and membrane structures that were not identifiable - these may be partially degraded mitochondria, compacted autophagosomal membranes or tubules or other structures. Moreover, YFP-LC3 in *TBC1D15*^*-/-*^ cells sometimes associates with small diameter membrane structures such as late endosomes localized very close to mitochondria. However, these may connect to isolation membranes out of plane of the EM section. We think the LC3-label seen by fluorescence microscopy is derived from all these different types of LC3-labeled membranes. Of note, immunoelectron microscopy data indicates that LC3-positive structures in *TBC1D15*^*-/-*^ cells are not aggregates or aggresomes of LC3 protein. We added these electron microscopy data to new Figure 2F and G and Figure 3B.

*2) The effect of deletion of FIS1 or TBC1D15 on mitophagy is modest. The authors only show that the level of TOMM20 is higher in* FIS1^-/-^
*and* TBC1D15^-/-^
*cells compared to wild-type cells at 40 h after valinomycin treatment. As this is one of the main conclusions of this study, it should be more vigorously investigated, for example by quantification of mtDNA and other protein markers*.

Using YFP-Parkin stably expressing cells, we could detect and quantify the mtDNA-encoded protein COXII by immunoblotting. We also quantified the decrease of the outer membrane protein TOMM20 and the matrix protein HSP60. After 40 hrs valinomycin treatment, degradation of not only TOMM20 but also HSP60 and COXII was retarded in *FIS1*^*-/-*^ and *TBC1D15*^*-/-*^ when compared to that in WT cells. Although this is not a major conclusion of our paper, these results indicate that loss of Fis1 or TBC1D15 affect the efficient clearance of damaged mitochondria by Parkin/PINK1 mediated mitophagy. We separated these mitophagy data (original Figure 2F and G) from the original Figure 2 and made a new Figure 4 (the original Figure 4 is now Figure 5).

*3) The role of Rab7 in mitophagy is also not convincing. Although there seems to be a significant difference in Figure 7C, the images in Figure 7B do not convincingly support this quantification results. In particular, at which step mitophagosome formation is blocked is not clearly demonstrated. Again, electron microscopy can distinguish whether membrane formation or fusion is inhibited at an early or late step, or recognition of mitochondria by autophagic membranes is impaired*.

We now provide low magnification images with 10-20 cells per field that show Rab7 knock down erases LC3-labeled accumulation and/or tubulation. The lower magnification images display a significant and clear difference of LC3 accumulation in *TBC1D15*^*-/-*^ cells (and also *FIS1*^*-/-*^ cells) dependent on Rab7 knock down. These data are now presented as new Figure 8–figure—figure supplement 1A. We hope that these data adequately support the quantification results in Figure 8C.

Although several published papers have reported that Rab7 is involved in fusion between lysosome and autophagosome, we see a more upstream activity of Rab7 function in mitophagy such as the relation of Rab7 to microtube-dependent movement that is not detected during starvation induced autophagy. We have revised the Discussion to make this more clear.

*4) The order of figures is not well structured and not easy to follow throughout the manuscript. For example,*
Figure 5
*is referred prior to*
Figure 1*, and parts of*
Figure 2
*are discussed later in the manuscript after*
Figure 3*. Please consider rearranging the figure to fit with the order of description in the text*.

We have separated the original Figure 2 (mitophagy assays) from Figure 2 and moved them to a new Figure 4 with new data (the original Figure 4 became now a new Figure 5). We also moved the original Figure 1 (mitochondrial morphology of *TBC1D17*^*-/-*^ and *TBC1D15/17 DKO* cells) to a new Figure 6–figure—figure supplement 2. We also separated the original Figure 2, with part moving into the new Figure 7 and Figure 8.

*5) The authors need to describe how they define each group in*
Figure 1, Figure 2, Figure 2—figure supplement 2, Figure 2—figure supplement 3, Figure 3, Figure 4, Figure 5 and Figure 7*. Especially, the criteria how the authors count diffuse, punctate, and aggregated LC3 should be clearly stated. The authors also need to state how many cells they counted for the statistical analyses*.

As the reviewers requested, we have now described the criteria of mitochondrial morphology and Parkin translocation in the figure legends. For the criteria of LC3 morphology, we made a new Figure (Figure 2—figure supplement 1) with specific image examples of diffuse, puntate and accumulated LC3. We counted over 100 cells in each of triplicate wells for LC3 morphology and over 50 cells each in triplicate wells for the other statistical analyses including mitochondrial morphology, starvation-induced autophagy, and rescue experiments. This information is now added to the figure legends.